# RWKV-7 "Goose" with Expressive Dynamic State Evolution

**Bo Peng** *
EleutherAI,
RWKV Project (Linux
Foundation AI & Data)
pengbo863@gmail.com

**Ruichong Zhang** *
Tsinghua University
zhangrc24@mails
.tsinghua.edu.cn

**Daniel Goldstein** *
EleutherAI,
Recursal AI
dan@recursal.ai

**Eric Alcaide**
EleutherAI
USI, IDSIA

**Xingjian Du**
University of
Rochester

**Haowen Hou**
Guangdong
Laboratory of
AI and Digital
Economy (SZ)

**Jiaju Lin**
Pennsylvania
State
University

**Jiaxing Liu**
Zhejiang
University

**Janna Lu**
Recursal AI,
George Mason
University

**William Merrill**
New York
University

**Guangyu Song**
EleutherAI
Tano Labs

**Kaifeng Tan**
Shenzhen
University

**Saiteja Utpala**
EleutherAI

**Nathan Wilce**
EleutherAI,
Recursal AI

**Johan S. Wind**
University
of Oslo

**Tianyi Wu**
Beijing Normal
University

**Daniel Wuttke**
EleutherAI,
Denigma

**Christian Zhou-Zheng**
EleutherAI

## Abstract

We present RWKV-7 "Goose", a new sequence modeling architecture with constant memory usage and constant inference time per token. Despite being trained on dramatically fewer tokens than other top models, our 2.9 billion parameter language model achieves a new 3B SoTA on multilingual tasks and matches the current 3B SoTA on English language downstream performance. RWKV-7 introduces a newly generalized formulation of the delta rule with vector-valued gating and in-context learning rates, as well as a relaxed value replacement rule. We show that RWKV-7 can perform state tracking and recognize all regular languages, while retaining parallelizability of training. This exceeds the capabilities of Transformers under standard complexity conjectures, which are limited to $\mathsf{TC}^0$. To demonstrate RWKV-7's language modeling capability, we also present an extended open source 3.1 trillion token multilingual corpus, and train four RWKV-7 models ranging from 0.19 billion to 2.9 billion parameters on this dataset. To foster openness, reproduction, and adoption, we release our models[1] and dataset component listing[2] on Hugging Face, and our training and inference code[3] on GitHub; all under the Apache 2.0 License.

## 1 Introduction

Autoregressive Transformers (Vaswani et al., 2017) have recently dominated sequence modeling tasks, enjoying excellent in-context processing and highly parallelizable training due to their use of softmax attention. However, softmax attention incurs quadratic computational complexity and memory usage with respect to sequence length due to its linearly expanding key-value cache. For

---

*Equal first authorship. Others listed alphabetically.

[1]Model weights at https://huggingface.co/RWKV

[2]Dataset components listed at https://huggingface.co/RWKV

[3]Source code at: https://github.com/RWKV/RWKV-LM

| Name | State Evolution | Scalars | LS | FD | DD | GE |
|---|---|---|---|---|---|---|
| RWKV-4 | $s_t = e^{-w} \odot s_{t-1} + e^{k_t} \odot v_t;$ $s'_t = e^{-w} \odot s'_{t-1} + e^{k_t}$ | | ✗ | ✓ | ✗ | ✗ |
| RetNet | $S_t = wS_{t-1} + v_t^T k_t$ | $w$ | ✓ | ✗ | ✗ | ✗ |
| RWKV-5 | $S_t = S_{t-1}\text{diag}(w) + v_t^T k_t$ | | ✓ | ✓ | ✗ | ✗ |
| Mamba | $S_t = S_{t-1} \odot \exp(-(w_t^T 1) \odot \exp(A)) + (w_t \odot v_t)^T k_t$ | | ✓ | ✓ | ✓ | ✗ |
| RWKV-6 & GLA | $S_t = S_{t-1}\text{diag}(w_t) + v_t^T k_t$ | | ✓ | ✓ | ✓ | ✗ |
| HGRN-2 | $S_t = S_{t-1}\text{diag}(w_t) + v_t^T(1 - w_t)$ | | ✓ | ✓ | ✓ | ✗ |
| Mamba-2 | $S_t = w_t S_{t-1} + v_t^T k_t$ | $w_t$ | ✓ | ✗ | ✓ | ✗ |
| TTT [a] | $S_t = S_{t-1} - a_t \nabla l(S_{t-1}, k_t, v_t)$ | $a$ | ✓ | ✗ | ✗ | ✓ |
| Longhorn | $S_t = S_{t-1} \odot (I - a_t^T k_t^2) + (a_t x_t)^T k_t$ | | ✓ | ✓ | ✓ | ✗ |
| Gated DeltaNet | $S_t = w_t S_{t-1}(I - a_t k_t^T k_t) + a_t v_t^T k_t$ | $w_t, a_t$ | ✓ | ✗ | ✓ | ✓ |
| Titans [a] | $M_t = (1 - \alpha_t)M_{t-1} + S_t$ $S_t = w_t S_{t-1} - a_t \nabla l(M_{t-1}, k_t, v_t)$ | $w_t, a_t$ | ✓ | ✗ | ✓ | ✓ |
| Generalized Δ Rule | $S_t = S_{t-1}(\text{diag}(w_t) + z_t^T b_t) + v_t^T k_t$ | | ✓ | ✓ | ✓ | ✓ |
| **RWKV-7 (ours)** | $S_t = S_{t-1}(\text{diag}(w_t) - \hat\kappa_t^T(a_t \odot \hat\kappa_t)) + v_t^T k_t$ | | ✓ | ✓ | ✓ | ✓ |

Table 1: Recent RNN architectures used for language modeling.
**LS** (Large State): matrix-valued states, or state size at least 4 times larger than the model dimension.
**FD** (Flexible Decay): the dimension of the decay term $w$ or $w_t$ is not smaller than the model dimension.
**DD** (Dynamic Dependence): the decay term $w_t$ is a function over the input $x_t$.
**GE** (Generalized Eigenvalue): evolution matrix admits eigenvalues outside of the interval $[0, 1]$.
[a] Shown with mini batch size 1 for simplicity.

short sequences, much of this cost can be covered by modern GPU parallelism techniques (Dao, 2023), but Transformer inference becomes increasingly costly as sequence lengths grow.

This limitation has inspired significant research into the design of recurrent neural network (RNN) architectures with compressive states that afford linear computational complexity and constant memory usage, while still allowing highly parallel training. Two of the most commonly proposed alternatives that satisfy these requirements are linear attention variant models (Katharopoulos et al., 2020b; Sun et al., 2023; Peng et al., 2024b; Yang et al., 2023a) and State Space Models (Gu & Dao, 2023). These architectures have grown more sophisticated, with many recent proposals incorporating some form of the delta rule, as embodied by parallelized DeltaNet (Schlag et al., 2021; Yang et al., 2024c). Such models have achieved impressive downstream performance results: since RWKV-4 (Peng et al., 2023), RNN models have shown increasing potential to rival Transformers when given equivalent model size and training compute, while dramatically reducing inference costs.

We present a new architecture, RWKV-7 "Goose", which generalizes the delta rule for use in sequence modeling. First, we add a vector-valued state gating mechanism, enhancing expressivity and providing implicit positional encoding. Second, we expand the in-context learning rate (see Section 2) from a scalar to become vector-valued, allowing the model to selectively replace state data on a channel-wise basis. Third, we decouple the keys at which the delta rule removes from and adds to the state. Finally, we place these innovations within a modified RWKV-6 architecture, inheriting important features such as token-shift, bonus, and a ReLU$^2$ feedforward network. We also introduce an expanded 3.1 trillion token RWKV World v3 corpus designed for enhanced English, code, and multilingual task performance. We use this architecture and corpus to train new state-of-the-art open-source language models, upgraded from preexisting RWKV-5/RWKV-6 checkpoints.

Our main contributions are as follows:

- The **RWKV-7 "Goose" architecture**, which dramatically improves downstream benchmark performance over RWKV-6 and demonstrates state-of-the-art multilingual performance at 3B scale and near SoTA English language performance, despite being trained on many fewer tokens than the top models in its class.

- The **RWKV World v3 public dataset**, comprised of 3.1 trillion tokens of publicly available multilingual data.
- Public release of four **pretrained RWKV-7 World v3 language models**, ranging from 0.19 to 2.9 billion parameters trained on 1.6 to 5.6 trillion tokens.
- Public release of three **pretrained RWKV-7 Pile language models**, using the GPT-NeoX tokenizer (Black et al., 2022), ranging from 0.17 to 1.47 billion parameters, useful for comparative study with other architectures.
- **Proofs** that the generalized delta rule employed in RWKV-7 can **solve problems outside of** $\mathsf{TC}^0$ under the widely held complexity conjecture that $\mathsf{TC}^0 \neq \mathsf{NC}^1$. This includes solving an $S_5$ state tracking problem known to be in $\mathsf{NC}^1$ using only a single layer, and recognizing all regular languages using only a constant number of layers.
- A **method for upgrading the RWKV architecture without pretraining from scratch**, producing increasingly competitive trained models at reduced computational expense.

## 2 Background

Linear attention's major advantage over softmax attention is that it can be formulated as a RNN with constant running time per token and constant memory usage (Katharopoulos et al., 2020a), while softmax attention takes $O(N)$ time per token and $O(N)$ memory with regard to sequence length. Despite this dramatic efficiency improvement, linear attention has its own significant drawbacks including a fixed-size state that is added to but never truly erased (Schlag et al., 2021; Han et al., 2024; Fan et al., 2025; Yang et al., 2024b).

**Delta Rule.** DeltaNet (Schlag et al., 2021) sidesteps the problem of additive state by partially replacing the value stored at the current key with the same amount of a new value, allowing the model to both take away old memories and add new ones on a per-key basis. It reformulates the state update as an explicit online learning problem where the goal is to retrieve the correct value as output for a given key as input. DeltaNet was the first to apply the foundational Error Correcting Delta Rule (Widrow et al., 1960) to key-value compressive states, akin to those stored in the RNN formulation of linear attention. This update rule is equivalent to a single step of stochastic gradient descent, training the state $S_t$ at test time to output the desired values $v_t$ for the keys $k_t$ as inputs using loss $\mathscr{L} = \frac{1}{2}\|(S_t k_t - v_t)\|^2$ and gradient $\frac{\partial \mathscr{L}}{\partial S} = Sk^\top k - v^\top k$, leading to a recurrent update formula of $S_t = S_{t-1}(I - ak_t^T k_t) + av_t^T k_t$, where $a$ is a scalar learning rate. The ideas behind this internal state update can be traced back to fast weights (Schmidhuber, 1992).

There has been significant recent interest in improvements to DeltaNet, in order to bring its efficiency and downstream performance in line with Transformers while still capturing the speed and memory benefits of Linear Attention. Parallelizing DeltaNet (Yang et al., 2024c) showed that DeltaNet used diagonal plus low-rank (DPLR) state evolution like S4 (Gu et al., 2022), and could be parallelized across the time dimension, creating a path to efficiently train such models. Our work further extends that parallelization to cover the generalized delta rule formulation introduced herein, as well as the specific formula of RWKV-7.

**Concurrent Work.** Concurrent work with our own has focused on architectural improvements beyond DeltaNet while still using the delta rule or variations thereof. Longhorn (Liu et al., 2024) employs an update rule that approximates a closed-form solution to a globally optimal update objective, applied on an otherwise unchanged Mamba architecture. Gated Delta Networks (Yang et al., 2024a) applies gating $w_t$ to the DeltaNet state, essentially multiplying the transition matrix by a data-dependent scalar per head. This combines the DeltaNet update rule with the scalar decay found in some modern RNNs like RetNet and Mamba-2. TTT (Test-Time Training) (Sun et al., 2024) and Titans (Behrouz et al., 2024) also both apply scalar decay, but eschew per-step gradient descent update rules in favor of a batched multi-timestep approach. Titans also adds momentum to the otherwise classical SGD update applied to the state.

Grazzi et al. (2024) demonstrated the potential for increased expressiveness that comes from allowing the state transition matrix to contain negative eigenvalues. We show a result signifi-

cantly beyond this, proving that RWKV-7 and our generalized delta rule can recognize all regular languages using only a small constant number of layers.[4]

## 3 Architecture

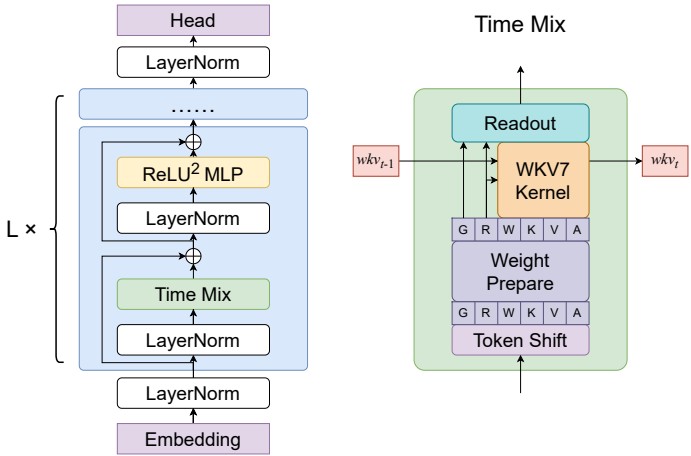

Figure 1: RWKV-7 overall architecture. See Appendix F for details.

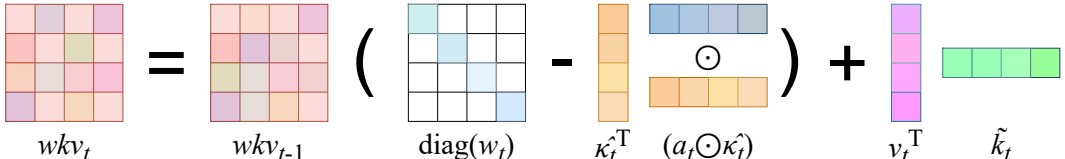

Figure 2: A simple illustration of the update mechanism of a single head of RWKV-7's state. Note that the actual state size is $64 \times 64$ per head, not $4 \times 4$.

Unlike the other work described above, RWKV-7 generalizes the delta update rule into an extended formula $S_t = S_{t-1}(\mathrm{diag}(w_t) + z_t^T b_t) + v_t^T k_t$ to increase expressivity (see Table 1). This is still a diagonal plus rank one update rule, which admits efficient forms of parallelization (Yang et al., 2024c). Here, $w_t$ is a more expressive data-dependent vector-valued decay, unlike the scalar decays featured in the other works previously described. Our use of $z_t$ and $b_t$ in this extended formula permits a flexible approach to state update, while retaining the important reduction in non-SRAM memory bandwidth usage that comes from using small data-dependent vectors instead of large matrices. One example of this flexibility is our use of different removal and replacement keys.

RWKV-7's extended delta rule replaces data-dependent vector-valued amounts of the state, allowing each key channel in the state to vary independently. We parameterize $z_t = -\hat{\kappa}_t$ and $b_t = \hat{\kappa}_t \odot a_t$ where $a_t$ has elements in $(0, 1)$. This keeps the update stable (see Appendix C), while maintaining increased expressivity. We show ablations for these improvements in Appendix M.2.

RWKV-7 has a non-diagonal and input-dependent transition matrix, allowing it to represent more complex functions than its predecessors. (Merrill et al., 2024) In fact, we demonstrate that RWKV-7 possesses expressive power surpassing that of $\mathsf{TC}^0$ under standard complexity conjectures and can recognize all regular languages with a constant number of layers. One new component of this power, not present in the original delta rule, is the ability to represent the "copy" state transition (Lemma 3). This is a key element in our proof. As a result, we have the following theorems:

**Theorem 1.** *RWKV-7 can solve a problem which is* $\mathsf{NC}^1$*-complete under* $\mathsf{AC}^0$ *reductions.*

**Theorem 2.** *For any regular language, there exists a 4-layer RWKV-7 model that recognizes it.*

---

[4]For now, we choose to allow only part of the range of possible negative eigenvalues in our pretrained large language models due to experimentally observed training instabilities.

See Appendix D for proofs and details.

We replace the main RWKV-6 (Peng et al., 2024b) diagonal transition matrix with our extended delta rule and make several other changes to the RWKV architecture, observing significant modeling improvements. These include updates to the channel mixing module and the token shift module. We remove the data dependency of token-shift and the receptance gating of channel mixing, both of which contribute to faster training and inference. We increase the use of low-rank projections to generate more of our intermediate calculations, striking a balance between the total number of model parameters, training and inference speed, and downstream performance.

## 4 Method

In this section, we use $D$ to denote the model dimension. Bold capital letters represent trainable matrices, and vectors without a subscript $t$ are trainable parameters. The first subscript denotes sequence position and second subscript denotes layer index, where necessary. We use the convention that all vectors are row vectors unless explicitly transposed, so all matrices operate on the right side, therefore $a^T b$ is an outer product and $ab^T$ is an inner one. We use the square subscript to denote a placeholder for variable names and use the $\prod$ sign for cumulative matrix multiplication. See Appendix G for a pseudocode implementation of these formulas.

### 4.1 Time Mixing

**Weight Preparation** Along the lines of (Peng et al., 2024b), we introduce the following notation templates for common operators in the model, using the square subscript to denote a variable:

$$\text{lerp}(a, b, x) = a + (b - a) \odot x, \tag{1}$$

$$\text{loramlp}_\square(f, x, \text{bias}) = f(x\boldsymbol{A}_\square)\boldsymbol{B}_\square + (\lambda_\square \text{ if bias else } 0), \tag{2}$$

Unless explicitly stated, all vectors appearing in this section are dimension $D$.

We extend the use of low-rank MLP (a 2-layer MLP with small hidden dimension compared to input and output), abbreviated as loramlp, to implement data dependency using minimal parameters.

The replacement key $\tilde{k}$, value $v$, decay $w$, removal key $\kappa$, in-context learning rate $a$, receptance $r$, and rwkv gate $g$ parameters are computed as follows (outputs annotated with $\triangleright$):

$$x_t^\square = \text{lerp}(x_t, x_{t-1}, \mu_\square) \quad \square \in \{r, k, v, d, a, g\}, \qquad \text{token shifted inputs} \tag{3}$$

$$a_t = \text{sigmoid}(\text{loramlp}_a(\text{Identity}, x_t^a, \text{bias=True})), \qquad \triangleright\text{in-context learning rate} \tag{4}$$

$$k_t = x_t^k \boldsymbol{W}_k, \qquad \text{key precursor} \tag{5}$$

$$\kappa_t = k_t \odot \xi, \qquad \triangleright\text{removal key} \tag{6}$$

$$\tilde{k}_t = k_t \odot \text{lerp}(1, a_t, \alpha), \qquad \triangleright\text{replacement key} \tag{7}$$

$$v_t = \text{sigmoid}(\text{loramlp}_v(\text{Identity}, x_t^v, \text{bias=True})), \qquad \text{value residual gate} \tag{8}$$

$$v'_{t,l} = x_t^v \boldsymbol{W}_v, \qquad \text{value precursor} \tag{9}$$

$$v_t = \begin{cases} v'_{t,0}, & \text{layer } l = 0 \\ \text{lerp}(v'_{t,0}, v'_{t,l}, v_t), & \text{layer } l \geq 1 \end{cases}, \qquad \triangleright\text{value} \tag{10}$$

$$d_t = \text{loramlp}_d(\tanh, x_t^d, \text{bias=True}), \qquad \text{decay precursor} \tag{11}$$

$$w_t = \exp(-e^{-0.5}\text{sigmoid}(d_t)), \qquad \triangleright\text{decay} \tag{12}$$

$$r_t = x_t^r \boldsymbol{W}_r, \qquad \triangleright\text{receptance} \tag{13}$$

$$g_t = \text{loramlp}_g(\text{sigmoid}, x_t^g, \text{bias=False}) \qquad \triangleright\text{rwkv gate} \tag{14}$$

$\xi$ is a learned parameter representing the removal key multiplier, which transforms the original key into a version to be removed from the state. In practice, $\xi$ lies in a range of approximately $[-5.3, 9.4]$.

$\alpha$ is a learned parameter representing the replacement rate booster, which adjusts the amount added back to the state after the transition matrix is applied.

Unlike $r, k$ and $v$ which are the main carriers of information, $g, d, v$ and $a$ act like gates which control the amount of information allowed to pass.

For comprehensive statistics of $\xi$, $\alpha$ and biases of $d_t$ observed in the released RWKV-7 model, including extremum values, mean measurements, and distribution trends, see Appendix O.

For the computation of $x_t^{\square}$, we removed data dependency of linear interpolation from RWKV-6 to improve training speed.

We adapted the idea of Value Residual Learning Zhou et al. (2024) for the computation of $v_t$, which has shown to improve the final language modeling loss. $v_t$ represents the value residual mix, which interpolates between the layer zero and current layer value precursors: $v_{t,0}$ and $v_{t,l}$.

We also updated the formula for computation of $w_t$, restricting all entries in $(\exp(-e^{-0.5}), 1)$ in favor of a smaller condition number for $\mathrm{diag}(w_t)$, which maintains better training stability, and was beneficial to accuracy of the backward pass.

The $\tilde{k}_t$ in the formula can be regarded as a "normalized key", a design to ensure that the state of $\boldsymbol{wkv}$ contains columns of $O(1)$ size. Normally, we expect $\tilde{k}_t = k_t \odot (1 - w_t)$, as employed in RWKV-6c (see Appendix F), so that $\boldsymbol{wkv}_t$ rows are linear interpolations between $\boldsymbol{wkv}_{t-1}$ and $v_t^T k_t$ controlled by $w_t$. However, to further enhance expressivity, we decide to decouple $w_t$ and $a_t$. We further decouple $a_t$ from the amount actually added to the state, allowing the replacement rate booster $\alpha$ to interpolate the amount added between the normal in-context learning rate and 1.0. Importantly, all of these modifications operate on a per-channel basis. The numerical range of RWKV-7's $\boldsymbol{wkv}$ entries are generally stable as in RWKV-6c, unlike RWKV-6, where entries of the states can accumulate to thousands (see Appendix N for a state visualization).

**The Weighted Key Value State Evolution**   After weight preparation, we reshape $(r, w, \tilde{k}, v, \kappa, a)_t$, splitting them to $h$ heads, with each head sized $D/h$. We always assume that $h$ is a factor of $D$ and heads are equally split. All operations in this section are shown per-head. Before mixing in the time dimension, $\kappa_t$ is normalized per head:

$$\hat{\kappa}_t = \kappa_t / \|\kappa_t\|_2 \tag{15}$$

The $\boldsymbol{wkv}$ (Weighted Key Value) is a multi-headed matrix-valued state of fast weights that undergoes dynamic evolution. The evolution of $\boldsymbol{wkv}$ is crucial for encoding context information by learning at test time to map keys to values. We start by defining the WKV time mixing as the recurrence relation

$$\boldsymbol{wkv}_0 = \boldsymbol{0}, \tag{16}$$

$$\boldsymbol{wkv}_t = \boldsymbol{wkv}_{t-1} \left( \mathrm{diag}(w_t) - \hat{\kappa}_t^T (a_t \odot \hat{\kappa}_t) \right) + v_t^T \cdot \tilde{k}_t \tag{17}$$

Compared to RWKV-5 and RWKV-6, the $\boldsymbol{wkv}$ in this paper is transposed to ensure consistency with RWKV-7's code. The $\boldsymbol{wkv}_t$ attention calculation can alternatively be written in a parallel manner:

$$\boldsymbol{wkv}_t = \sum_{i=1}^{t} \left( v_i^T \tilde{k}_i \prod_{j=i+1}^{t} \left( \mathrm{diag}(w_j) - \hat{\kappa}_j^T (a_j \odot \hat{\kappa}_j) \right) \right) \in \mathbb{R}^{(D/h) \times (D/h)} \tag{18}$$

The recurrent transition design has parallels with Schlag et al. (2021), but crucially the transition matrix

$$G_t = \mathrm{diag}(w_t) - \hat{\kappa}_t^T (a_t \odot \hat{\kappa}_t) = \left( \boldsymbol{I} - \hat{\kappa}_t^T (\frac{a_t}{w_t} \odot \hat{\kappa}_t) \right) \mathrm{diag}(w_t) \approx \left( \boldsymbol{I} - 2\hat{\kappa}_t^T \hat{\kappa}_t \right) \mathrm{diag}(w_t) \tag{19}$$

is no longer a Householder matrix but a scaled approximation of it, as $\hat{\kappa}_t \neq \frac{a_t}{w_t} \hat{\kappa}_t$. This mimics a Householder matrix but with expanded dynamics, while still having all eigenvalues in a stable range of $[-1, 1]$ and allows the network to decay information in all subspaces if necessary. It contrasts with the case of a Householder-like matrix with learning rate $(I - a v^T v)$, $a \in [0, 1]$, as used in Schlag et al. (2021); Yang et al. (2024c) where all eigenvalues are one except for the last one corresponding to $1 - a$. Given these properties, we refer to $w_t$ as "in-context weight decay" and to $a_t$ as "in-context learning rate" (ICLR). The RWKV-7 transition matrix, therefore, allows for both dynamic state evolution and approximation to a forget gate at the same time. See Appendix C for the details on the eigenvalue of the transition matrix, and when the transition matrix is guaranteed to be stable.

**WKV Bonus and Output**    All operations in this section are shown per-head unless otherwise specified. Receptance, which acts like the query found in transformers, is applied to the WKV state, and the result is normalized. An added bonus, the amount of which is weighted by $\rho$, allows the model to place extra attention on the current shifted input token without requiring it to store that token in the state.

$$u_t = \left(r_t \cdot (\rho \odot \tilde{k}_t)^T\right) v_t \qquad\qquad \text{bonus} \qquad\qquad (20)$$

$$p_t = \text{LayerNorm}(r_t \boldsymbol{wk}\boldsymbol{v}_t^T) + u_t \qquad\qquad \triangleright \text{attention result} \qquad\qquad (21)$$

Finally, the heads are recombined via reshaping so that $p_t \in \mathbb{R}^D$, gated, and transformed into the output as follows:

$$o_t = (g_t \odot p_t)\boldsymbol{W}_o \in \mathbb{R}^D \qquad\qquad (22)$$

## 4.2   MLP

The MLP module of RWKV-7 is no longer identical to the Channel Mixing module of previous RWKV-4,5,6 architectures (Peng et al., 2024b). We remove the gating matrix $\boldsymbol{W}_r$, making it a two-layer MLP. In compensation for the removed gating parameters to satisfy the equi-parameter condition, we set the hidden dimension to be 4 times the size of model dimension.

$$k'_t = \text{lerp}(x'_t, x'_{t-1}, \mu'_k)\boldsymbol{W}_{k'} \in \mathbb{R}^{4D} \qquad\qquad (23)$$

$$o'_t = \text{ReLU}(k'_t)^2 \boldsymbol{W}_{v'} \in \mathbb{R}^D \qquad\qquad (24)$$

## 5   RWKV World v3 Dataset

We train our models on the new **RWKV World v3 Dataset**, a new multilingual 3.119 trillion token dataset drawn from a wide variety of publicly available data sources. This dataset aims to help close the gap with the amount of data used to train modern LLMs, which may consume as many as 15 - 18 trillion tokens (Qwen et al., 2025; Grattafiori et al., 2024). We select the data to approximate the distribution of our previous World datasets, including English, multilingual, and code, while slightly enhancing Chinese novels. We describe the composition of our dataset in Appendix B.

## 6   Pretrained Models

We have pretrained and publicly released seven Apache 2.0 licensed RWKV-7 models:

1. Trained on Pile: **RWKV7-Pile** of sizes **0.1B, 0.4B, and 1.4B**
2. Trained on RWKV World V3: **RWKV7-World-3** of sizes **0.1B, 0.4B, 1.5B, and 2.9B**

Due to compute budget constraints, we used a new method to convert and train these models from upgraded pre-existing RWKV-5 and 6 checkpoints. See Appendix E for details.

## 7   Experiments

**Speed and Memory Usage**    We test training speed and memory usage of RWKV-7, RWKV-6, and Flash Attention v3 (Shah et al., 2024) kernels on H100 SXM. Due to linear scaling, RWKV-7 surpasses Flash Attention's speed around 4096 sequence length. The new RWKV-7 kernel is also about 3x faster than the official RWKV-6 kernel. See Figure 3 and Appendix I for details.

**LM Evaluation Harness Benchmarks**    We evaluated RWKV-7 alongside several new open models which are state-of-the-art in their parameter count ranges on a series of common English and multilingual benchmarks using *LM Evaluation Harness* v0.4.8 (Gao et al., 2023) in fp32, as shown in Figures 4 and 7 and Tables 2 and 3.

The RWKV-7-World models establish a new multilingual state-of-the-art for their size, expanding upon RWKV-6-World models already strong capabilities on multilingual benchmarks and significantly outperforming SmolLM2 (Allal et al., 2025), Llama-3.2 (Grattafiori et al., 2024), and

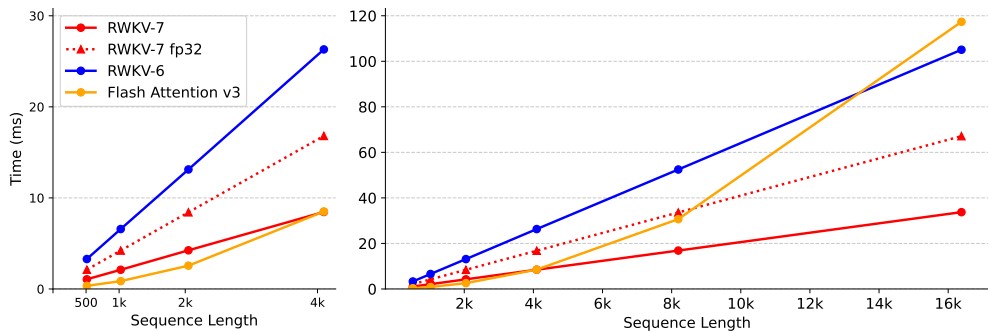

Figure 3: Kernel forward+backward time vs. sequence length (H100)

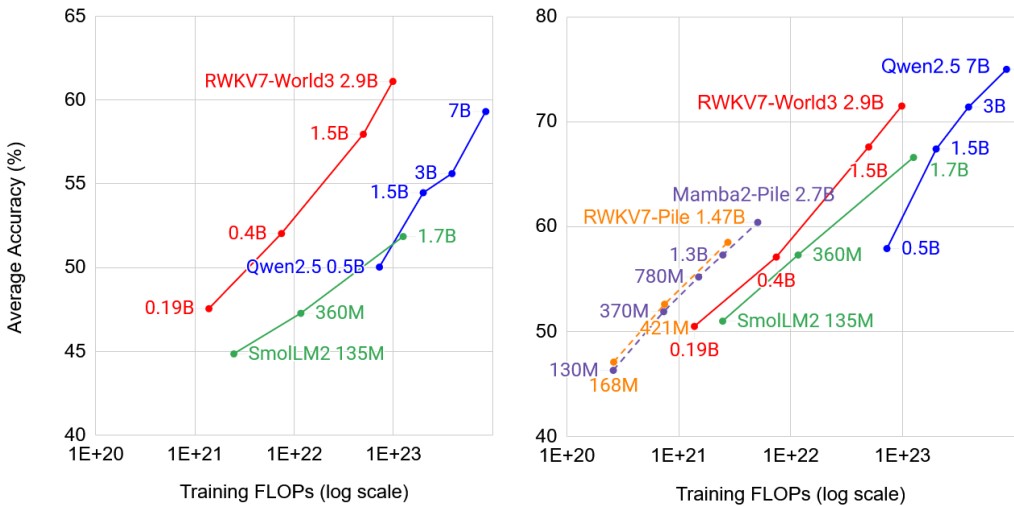

Figure 4: FLOPS vs. average benchmark accuracy for multilingual (left) and English (right). Multilingual benchmarks omit Pile models due to a lack of multilingual training data in the Pile.

Qwen-2.5 (Qwen et al., 2025). We find that RWKV-7 is generally able to match the English performance of Qwen2.5 despite being trained on less than one third as many training tokens. In Figure 4 we plot FLOPs used for training versus average accuracy across the same sets of common benchmarks, demonstrating a very dramatic multilingual Pareto improvement versus the transformer models. We theorize that the difference would be even greater were we less compute constrained and able to train from scratch, instead of from pretrained checkpoints of earlier RWKV versions.

**Mechanistic Architecture Design** We evaluate RWKV-7 on the Mechanistic Architecture Design (MAD) benchmark (Poli et al., 2024), a suite of synthetic token manipulation tasks designed to probe architectural capabilities in sequence modeling, as shown in Table 4.

RWKV-7 achieves the highest average score across all six tasks, outperforming previous architectures. It demonstrates perfect accuracy on In-Context and Noisy Recall tasks, matching DeltaNet while setting a new state-of-the-art for Fuzzy Recall. RWKV-7 also shows strong performance in memorization and selective copying, suggesting effective combination of attention-based and recurrent model strengths.

**Recent Internet Data Evaluation** Despite careful data cleaning, benchmark data leakage remains a challenge, potentially compromising the validity of these evaluations. To complement traditional benchmarks, we evaluated RWKV-7 and other leading open-source models using temporally novel internet data, from after the models' training periods, removing data leakage concerns. See Table 5 and Appendix J.3 for full discussion and results.

| Model (Name) | Tokens (T) | lmb.o acc↑ | hella acc_n↑ | piqa acc↑ | arcE acc↑ | arcC acc↑ | glue [a]acc↑ | WG acc↑ | sciq acc↑ | mmlu acc↑ | avg acc↑ |
|---|---|---|---|---|---|---|---|---|---|---|---|
| RWKV5-World1-0.1B | 0.6 | 38.4 | 31.9 | 61.4 | 44.2 | 19.9 | 45.5 | 52.9 | 76.3 | 23.1 | 43.7 |
| **SmolLM2-135M** | 2.0 | 42.9 | **43.1** | **68.4** | **64.4** | **28.1** | **49.0** | **53.0** | 84.0 | **25.8** | **51.0** |
| RWKV7-World2.8-0.1B | 1.6 | **48.1** | 42.1 | 67.3 | 59.3 | 25.5 | 48.1 | 52.7 | **86.3** | 25.4 | 50.5 |
| RWKV5-World2-0.4B | 1.1 | 54.0 | 40.9 | 66.5 | 54.0 | 24.0 | 50.0 | 53.2 | 86.9 | 23.8 | 50.4 |
| SmolLM2-360M | 4.0 | 53.8 | 56.4 | 72.1 | **70.4** | **36.5** | **50.7** | 59.0 | 91.2 | 26.3 | 57.4 |
| **Qwen2.5-0.5B** | 18.0 | 52.5 | 52.1 | 70.2 | 64.6 | 29.5 | 54.7 | 56.4 | **93.1** | **47.8** | **57.9** |
| RWKV7-World2.9-0.4B | 3.1 | **58.6** | **56.8** | **72.9** | 68.7 | 31.9 | 49.4 | **59.9** | 89.7 | 26.1 | 57.1 |
| RWKV6-World2.1-1.6B | 2.5 | 67.4 | 61.1 | 74.4 | 64.3 | 31.0 | 51.0 | 60.7 | 89.5 | 25.1 | 58.3 |
| Llama3.2-1B | [b]15.0 | 63.0 | 63.7 | 74.5 | 65.5 | 31.3 | 49.7 | 60.7 | 91.4 | 32.1 | 59.1 |
| SmolLM2-1.7B | 11.0 | 67.7 | **71.5** | **77.0** | 77.7 | **44.7** | 51.5 | 66.1 | 93.3 | 50.3 | 66.6 |
| **Qwen2.5-1.5B** | 18.0 | 63.0 | 67.7 | 75.8 | 75.5 | 41.2 | **65.0** | 63.4 | 94.2 | **61.0** | 67.4 |
| **RWKV7-World3-1.5B** | 5.6 | **69.5** | 70.8 | 77.1 | **78.1** | 44.5 | 62.4 | **68.2** | **94.3** | 43.3 | **67.6** |
| RWKV6-World2.1-3B | 2.5 | 71.7 | 68.4 | 76.4 | 71.2 | 35.6 | 56.3 | 66.3 | 92.2 | 28.3 | 62.9 |
| Llama3.2-3B | [b]15.0 | 70.5 | 73.6 | 76.7 | 74.5 | 42.2 | 50.7 | 69.9 | 95.7 | 56.5 | 67.8 |
| **Qwen2.5-3B** | 18.0 | 67.1 | 73.5 | 78.6 | 77.4 | 45.0 | **70.2** | 68.5 | **96.2** | **65.7** | 71.4 |
| **RWKV7-World3-2.9B** | 5.6 | **73.4** | **76.4** | **79.7** | **81.0** | **48.7** | 61.8 | **72.8** | 95.0 | 55.0 | **71.5** |

[a] **glue** is the average accuracy of 8 subtasks: **mnli**, **mnli_mismatch**, **mrpc**, **qnli**, **qqp**, **rte**, **sst2** and **wnli**

[b] Llama3.2-1B and 3B were pruned and distilled from Llama3.1-8B (Grattafiori et al., 2024)

Table 2: English focused benchmarks, including LAMBADA (**lmb.o**) (Paperno et al., 2016), Hellswag (**hella**) (Hampel, 1974), PIQA (Bisk et al., 2020), AI2 ARC (**arcE**, **arcC**) (Bhakthavatsalam et al., 2021), GLUE (Wang et al., 2018), Winogrande (**WG**) (Sakaguchi et al., 2021), SciQ (Welbl et al., 2017), MMLU (Hendrycks et al., 2021). MMLU 5-shot, others 0-shot.

| Model (Name) | Tokens (T) | lmb.m [a]ppl↓ | lmb.m acc↑ | pawsx acc↑ | xcopa acc↑ | xnli acc↑ | xsClz acc↑ | xwin acc↑ | avg acc↑ |
|---|---|---|---|---|---|---|---|---|---|
| RWKV5-World1-0.1B | 0.6 | 270 | 22.0 | 48.6 | 53.0 | 36.1 | 51.7 | 59.5 | 45.1 |
| SmolLM2-135M | 2.0 | 1514 | 18.6 | **51.2** | 52.2 | 34.9 | 50.6 | 61.7 | 44.9 |
| **RWKV7-0.1B** | 1.6 | **114** | **31.6** | 46.1 | **53.3** | **37.6** | **52.6** | **64.1** | **47.5** |
| RWKV5-World2-0.4B | 1.1 | 66 | 36.8 | 49.5 | 54.0 | 38.5 | 54.1 | 65.6 | 49.8 |
| SmolLM2-360M | 4.0 | 389 | 25.8 | 51.4 | 51.7 | 36.0 | 51.2 | 67.8 | 47.3 |
| Qwen2.5-0.5B | 18.0 | 108 | 32.9 | **52.6** | 54.4 | 38.6 | 53.9 | 67.8 | 50.0 |
| **RWKV7-World3-0.4B** | 3.1 | **52** | **39.6** | 48.7 | **55.4** | **40.3** | **55.3** | **72.9** | **52.0** |
| RWKV6-World2.1-1.6B | 2.5 | 28 | 47.2 | 52.5 | 58.1 | 41.4 | 58.2 | 76.5 | 55.7 |
| Llama3.2-1B | [b]15.0 | 52 | 39.0 | 53.9 | 55.3 | 41.2 | 56.6 | 72.2 | 53.0 |
| SmolLM2-1.7B | 11.0 | 85 | 37.1 | **56.5** | 53.1 | 38.1 | 54.1 | 72.8 | 52.0 |
| Qwen2.5-1.5B | 18.0 | 49 | 40.0 | 55.3 | 57.4 | 40.6 | 57.7 | 75.8 | 54.5 |
| **RWKV7-World3-1.5B** | 5.6 | **25** | **48.4** | 54.8 | **59.7** | **43.7** | **61.4** | **79.8** | **58.0** |
| RWKV6-World2.1-3B | 2.5 | 21 | 51.0 | 53.4 | 60.2 | 42.7 | 61.3 | 78.8 | 57.9 |
| Llama3.2-3B | [b]15.0 | 30 | 45.9 | **59.9** | 58.5 | 44.2 | 60.6 | 79.2 | 58.1 |
| Qwen2.5-3B | 18.0 | 36 | 43.5 | 53.3 | 59.0 | 38.5 | 59.6 | 79.8 | 55.6 |
| **RWKV7-World3-2.9B** | 5.6 | **18** | **52.9** | 58.2 | **63.1** | **45.4** | **64.7** | **82.4** | **61.1** |

[a] The perplexity is the geometric mean, rather than arithmetic average, across 5 languages

[b] Llama3.2-1B and 3B were pruned and distilled from Llama3.1-8B (Grattafiori et al., 2024)

Table 3: Multilingual benchmarks, including LAMBADA Multilingual (**lmb.m**) (Gao et al., 2023), XCOPA (Ponti et al., 2020), XNLI (Conneau et al., 2018), XStoryCloze (**xsClz**) (Lin et al., 2022), xWinogrande (**xwin**) (Tikhonov & Ryabinin, 2021). All 0-shot.

**Additional Evaluations**   Please see Appendices J, K, L, and M for extended evaluations, multi-modal experiments, and ablations.

# 8   Conclusions

We introduced RWKV-7, a novel RNN architecture that achieves state-of-the-art performance for its size across a wide range of benchmarks, rivaling even highly optimized models such as Qwen2.5 despite being trained on many fewer tokens. As an RNN, RWKV-7 maintains high parameter

| Model | Compress | Fuzzy Recall | In-Context Recall | Memorize | Noisy Recall | Selective Copy | Avg |
|---|---|---|---|---|---|---|---|
| RWKV-7 | 44.5 | **43.2** | **100** | 89.1 | **100** | 98.8 | **79.3** |
| Transformer | 51.6 | 29.8 | 94.1 | 85.2 | 86.8 | 99.6 | 74.5 |
| Multihead Hyena | 44.8 | 14.4 | 99.0 | 89.4 | 98.6 | 93.0 | 73.2 |
| DeltaNet | 42.2 | 35.7 | **100** | 52.8 | **100** | **100** | 71.8 |
| Mamba | **52.7** | 6.7 | 90.4 | **89.5** | 90.1 | 86.3 | 69.3 |
| Hyena | 45.2 | 7.9 | 81.7 | 89.5 | 78.8 | 93.1 | 66.0 |
| GLA | 38.8 | 6.9 | 80.8 | 63.3 | 81.6 | 88.6 | 60.0 |

Results for comparison models from Yang et al. (2024c)

Table 4: Results on the MAD benchmark

| Model | arXiv CS ↓ | arXiv Phys. ↓ | Github Python ↓ | Github C++ ↓ | AO3 Eng ↓ | BBC news ↓ | Wiki Eng ↓ | average ↓ |
|---|---|---|---|---|---|---|---|---|
| **Qwen2.5-1.5B** | **8.12** | 8.65 | **4.42** | **4.40** | 11.76 | 9.58 | 9.49 | **8.06** |
| RWKV-7 1.5B | 8.25 | 8.77 | 5.57 | 5.29 | **10.93** | **9.34** | **8.97** | 8.16 |
| Llama-3.2-1B | 8.37 | 8.76 | 5.18 | 5.16 | 11.69 | **9.34** | 9.07 | 8.23 |
| SmolLM2-1.7B | 8.38 | 9.04 | 5.17 | 4.94 | 11.20 | 9.40 | 9.46 | 8.23 |

Table 5: Compression rate of LLMs on post January 2025 data sources

efficiency, linear time complexity, and constant memory usage, offering a compelling alternative to transformers.

**Limitations**  Despite its strengths, the architecture and models have some limitations, including prompt sensitivity, kernel precision limits, and a lack of full instruct and alignment tuning. We also believe that downstream performance has been restricted by our access to less training compute than other top models. See Appendix Q for a detailed discussion.

**Future Work**  We plan to train larger models, investigate training speed improvements, and train reasoning models that can take advantage of increased inference time compute. Please see Appendix R for details.

## Acknowledgments

We extend our gratitude to Shenzhen Yuanshi Intelligent Co. Ltd. and Shanghai Yuanwo Intelligent Co. Ltd. for providing computational resources and their dedication to promoting and commercializing RWKV. We thank Featherless AI for their extensive experimentation with the RWKV architecture and their contributions to this paper. We are grateful to the members of the RWKV and EleutherAI Discord communities for their collaborative efforts in extending the applicability of RWKV to diverse domains. We extend a special thank you to Stella Biderman, Songlin Yang and Yu Zhang. Eric Alcaide acknowledges support for this work from the Swiss State Secretariat for Education, Research and Innovation (SERI) under contract number 23.00421.

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

# A  Author Contributions

**Bo Peng**  Original RWKV-7 ideas, original code, performance optimizations, original experiments, dataset composition, and trained models from 0.1B to 2.9B.

**Ruichong Zhang**  Personnel organization, wrote Sections 3, 4, 7, J.2, 8 and Appendices C, E, H, N, M.1, P, Figures 1, 2, 5, 6, 15, 16 and Tables 1, 2, 3, 14, 9, 11, 12, 13, 19, 21. Additional contributions on implementing RWKV-7 for Flash-linear-attention and converting RWKV-7 models on HuggingFace.

**Daniel Goldstein**  Manuscript organization, initial draft sections 1, 2, 3, 4, 5, FLOPS portion of 7, figures 4 and 7, and appendices B, F, G. Proofreading and revisions of full manuscript. Oversaw and chose experiments for Appendix M.2 and for pass-key in subsection J.4. Assistance with Appendix D revisions and ideas. Developed and tested initial RWKV-7 Hugging Face model code.

**Eric Alcaide**  Section 3, validation of CUDA kernels for scalable training, and manuscript proofreading.

**Xingjian Du**  Experiments and writing of audio modeling in Appendix K.2

**Haowen Hou**  Wrote Section K.1, covering architectural design, coding, model training, experimental evaluation, as well as figure (Figure 12), table (Table 17), and text writing.

**Jiaju Lin**  Experiments and writing audio modeling in Appendix K.2

**Jiaxing Liu**  Experiments and writing audio modeling in Appendix K.2

**Janna Lu**  Needle-in-haystack evaluations for Section J.4, experiments for figure 4, figure 7, figure 10, table 2, and table 19. Edits to Section 1, G and abstract.

**William Merrill**  Developmental discussion, proofreading and revisions for Appendix D.

**Guangyu Song**  Section 7. Experiments for 7

**Kaifeng Tan**  Section 7, Figures 8, 9. Appendix L on board game modeling with Othello/Reversi, including training data design, model implementation, experiments, and result analysis.

**Saiteja Utpala**  Section J.5 and state tracking experiments for Figure 11.

**Nathan Wilce**  Extended context-length dataset development and extended-length model training. Description of extended context-length dataset in J.4

**Johan S. Wind**  Main author of RWKV-7 CUDA kernel implementations. Experiments for Figure 3. Section 7 and Appendices I, D. Contributions to Appendix C.

**Tianyi Wu**  Appendix D and Appendix O. Contributions to section 4 and Appendix C (proofreading and revisions).

**Daniel Wuttke**  Constructed an itemized table of v2-v3 world datasets. Proofreading and revision of the manuscript. Contributed to Abstract, Sections 1 and B. Contributed to Tables 6, 7, 8, and 12 Performed evaluations of RWKV-7 world models and reference base models (Table 2 and 3).

**Christian Zhou-Zheng**  Experiments and writing for Appendix M.2 and Table 20. Contributions to Sections 1 and 2. Proofreading and revisions of full manuscript.

## B   Training Dataset Details

The RWKV World v3 corpus builds upon the RWKV World v2 corpus (Peng et al., 2024b) in two steps that we describe separately here for the purposes of reproducibility for the Goose training runs: World v2.1 adds the entries listed in Table 6 to sum to a total of approximately 1.4 trillion RWKV World Tokenizer tokens. World v3 adds more entries, listed in Table 7, to sum to a total of approximately 3.1 trillion tokens. In the combined corpora, all tokens are given equal weighting unless otherwise noted.

| Dataset | Domain | Dataset | Domain |
|---|---|---|---|
| slimpajama C4 | Web | Llama-3-Magpie-Pro-1M-v0.1 | Align |
| dolma v1.6 (reddit only)[a] | Forums | Magpie-Pro-MT-300K-v0.1 | Align |
| glaive-code-assistant-v3 | Code | Magpie-Air-MT-300K-v0.1 | Align |
| m-a-p_Code-Feedback | Code | Magpie-Qwen2-Pro-1M-v0.1 | Align |
| cosmopedia-v0.1 | Synthetic | Magpie-Phi3-Pro-300K-Filtered- | Align |
| SystemChat-2.0 | Instruct | v1 | |
| Tess-v1.5 | Instruct | Magpie-Gemma2-Pro-200K- | Align |
| UltraInteract_sft | Instruct | Filtered-v0.1 | |

Table 6: Components added into the RWKV World v2.1 dataset, their source links, and their domains.
[a] We added only the reddit datasets from dolma v1.6
[b] DM_math as part of The Pile was in World v2 but missed being mentioned explicitly in (Peng et al., 2024b)

| Dataset | Domain | Dataset | Domain |
|---|---|---|---|
| REMOVED slimpajama parts[a] | Web | StarCoder[c] | Code |
| dclm-baseline-10-of-10[b] | Web | python-edu | Code |
| ccnews | Web | cosmopedia-v0.2 | Synthetic |
| fineweb-edu | Web Edu | WebInstructSub | Forums |
| TemplateGSM | Math | Buzz-v1.2 | Instruct |
| open-web-math | Math | SKGInstruct | Instruct |
| algebraic-stack | Math | FLAN | Instruct |

Table 7: Components added into the RWKV World v3 dataset, their source links, and their domains.
[a] We removed the CC and C4 components of SlimPajama from the corpus for World v3
[b] For DCLM-baseline, we include only global-shard_10_of_10
[c] For StarCoder, we now include all datasets, instead of just those datasets with at least 10 stars

| | |
|---|---|
| SlimPajama | Soboleva et al. (2023) |
| StarCoder | Li et al. (2023a) |
| Cosmopedia | Ben Allal et al. (2024a) |
| Dolma | Soldaini et al. (2024) |
| UltraInteract | Yuan et al. (2024) |
| Magpie | Xu et al. (2024) |
| FineWeb | Lozhkov et al. (2024) |
| DataComp LM (DCLM) | Li et al. (2024a) |
| WebInstructSub | Yue et al. (2024) |
| StructLM | Zhuang et al. (2024) |
| TemplateGSM | Zhang et al. (2024) |
| SmolLM Corpus | Ben Allal et al. (2024b) |
| FLAN | Wei et al. (2021) |
| OpenWebMath | Paster et al. (2023) |
| Algebraic-Stack | Azerbayev et al. (2024) |

Table 8: RWKV World v3 dataset component citations

Most of the component data sources for the RWKV World v3 dataset are used intact, with no up- or down-sampling done so that all tokens are given equal weighting. Some sub-sampling is done for

| Category | Tokens (B) |
|---|---|
| Web | 1945.2 |
| Books | 337.2 |
| Code | 258.4 |
| Science & Wiki | 222.7 |
| Fiction | 192.6 |
| Chat & QA & Instruction | 110.0 |
| Math | 32.3 |
| Law & Government | 19.0 |
| Poetry & Lyrics | 1.7 |
| Total | 3119.2 |

Table 9: RWKV World v3 dataset categories

over-represented languages within a few data sources in the original World v2 corpus. All newly added tokens in v2.1 and v3 are given equal weighting.

## C  Transition Matrix Eigenvalues and Stability

We are interested in all the eigenvalues of the transition matrix $A_t = \text{diag}(w_t) - \hat{\kappa}_t^T (a_t \odot \hat{\kappa}_t)$, and when it can be stable. We drop the subscript $t$ for statements which hold at all timesteps, to avoid clutter.

**Theorem 3.** *Let $A = \text{diag}(w) - c\hat{\kappa}^T (a \odot \hat{\kappa}) \in M_m(\mathbb{R})$ be a matrix, where all entries of $w$ belong to $(u, 1)$, where $u = \exp(-e^{-1/2}) = 0.5452\cdots$ is the clamping lower bound. Also, all entries of $a$ are located in $(0, 1)$, and $\hat{\kappa}$ is a unit row vector. When $c \in (0, 1 + u)$, the following holds:*

1. *The matrix $A$ is similar to a symmetric matrix, hence similar to a diagonal matrix.*
2. *All eigenvalues of $A$ lie in the interval $(-1, 1)$.*
3. *The matrix $A$ admits at most one negative eigenvalue.*
4. *If further assumed that $a_t$ is time-independent, then the update formula is guaranteed to be stable, i.e. there exists a time-independent constant $K$ such that*

$$\left\| \prod_{t=1}^{T} A_t \right\|_2 \leq K,$$

*where $\|\cdot\|_2$ denotes the spectral norm.*

*This hyperparameter $c$ in the formula can be regarded as a "Global ICLR Multiplier". This hyperparameter $c$ is set to $1$ in the current implementations of RWKV-7 language modeling.*

*Proof.*

1. We notice that

$$A = \text{diag}(w) - c\hat{\kappa}^T (a \odot \hat{\kappa}) = \text{diag}(w) - c\hat{\kappa}^T \hat{\kappa} \, \text{diag}(a).$$

The matrix $A$ itself is not necessarily a symmetric matrix. However, we can use the fact that $\text{diag}(a)$ is positive definite, so we can compute its square root. This allows us to rewrite

$$\text{diag}(a)^{1/2} A \, \text{diag}(a)^{-1/2} = \text{diag}(a)^{1/2} \text{diag}(w) \text{diag}(a)^{-1/2} - c \, \text{diag}(a)^{1/2} \hat{\kappa}^T \hat{\kappa} \, \text{diag}(a) \text{diag}(a)^{-1/2}$$

$$= \text{diag}(w) - c \left( \hat{\kappa} \text{diag}(a)^{1/2} \right)^T \left( \hat{\kappa} \text{diag}(a)^{1/2} \right),$$

which is a symmetric matrix. We denote this matrix by $B$. It has exactly the same eigenvalues as $A$, since it is formed by a similarity transformation of $A$.

2. Since $B = \mathrm{diag}(a)^{1/2} A \mathrm{diag}(a)^{-1/2}$ is symmetric, all eigenvalues of $B$ (and hence $A$) are real and located on the interval of

$$\left[ \min_{|\hat{s}|=1} \hat{s} B \hat{s}^T, \ \max_{|\hat{s}|=1} \hat{s} B \hat{s}^T \right].$$

It suffices to show that $\hat{s} B \hat{s}^T \in (-1, 1)$. This can be proved via direct expansion of $B$ by

$$\hat{s} B \hat{s}^T = \hat{s} \mathrm{diag}(w) \hat{s}^T - c \hat{s} (\hat{\kappa} \mathrm{diag}(a)^{1/2})^T (\hat{\kappa} \mathrm{diag}(a)^{1/2}) \hat{s}^T$$

$$\geq u \hat{s} \hat{s}^T - c \left| (\hat{\kappa} \mathrm{diag}(a)^{1/2}) \hat{s}^T \right|^2$$

$$\geq u - c$$

$$> -1.$$

Similarly,

$$\hat{s} B \hat{s}^T = \hat{s} \mathrm{diag}(w) \hat{s}^T - c \hat{s} (\hat{\kappa} \mathrm{diag}(a)^{1/2})^T (\hat{\kappa} \mathrm{diag}(a)^{1/2}) \hat{s}^T$$

$$< \hat{s} \hat{s}^T - c \left| (\hat{\kappa} \mathrm{diag}(a)^{1/2}) \hat{s}^T \right|^2$$

$$\leq \hat{s} \hat{s}^T$$

$$= 1,$$

which completes this part.

3. Recall $B = \mathrm{diag}(w) - c \left( \hat{\kappa} \, \mathrm{diag}(a)^{1/2} \right)^T \left( \hat{\kappa} \, \mathrm{diag}(a)^{1/2} \right)$. Then $B$ is congruent with

$$\hat{B} = \mathrm{diag}(w)^{-1/2} B \, \mathrm{diag}(w)^{-1/2} = I - u^T u,$$

where $u = \sqrt{c} \, \hat{\kappa} \, \mathrm{diag}(a)^{1/2} \mathrm{diag}(w)^{-1/2}$.

Clearly, $\hat{B}$ has at most one negative eigenvalue with value $1 - \|u\|^2$, and all other eigenvalues equal to 1. By Sylvester's law of inertia (Horn & Johnson, 2012), congruency preserves the number of negative eigenvalues. Hence, $B$ also has at most one negative eigenvalue.

4. We drop the subscript in the time-invariant $a_t$.

Since $B_t = \mathrm{diag}(a)^{1/2} A_t \mathrm{diag}(a)^{-1/2}$ is symmetric, the spectral norm of $B_t$ is equal to the largest absolute value for eigenvalues of $B_t$. We proved previously that the eigenvalues lie in $(-1, 1)$, so $\|B_t\|_2 \leq 1$. That is, $B_t$ is a contraction matrix.

Furthermore, we have

$$\prod_{t=1}^{T} A_t = \prod_{i=t}^{T} \mathrm{diag}(a)^{-1/2} B_t \mathrm{diag}(a)^{1/2}$$

$$= \mathrm{diag}(a)^{-1/2} \left( \prod_{i=1}^{t} B_t \right) \mathrm{diag}(a)^{1/2}.$$

Then

$$\left\| \prod_{t=1}^{T} A_t \right\|_2 \leq \left\| \mathrm{diag}(a)^{-1/2} \right\|_2 \left\| \prod_{t=1}^{T} B_t \right\|_2 \left\| \mathrm{diag}(a)^{1/2} \right\|_2$$

$$\leq \min(a)^{-1/2} \cdot 1 \cdot 1.$$

We can set $K = \min(a)^{-1/2} < \infty$, which is time-independent.

$\square$

While our stability proof only holds for time-independent $a_t$, we do not observe any problems with time-varying $a_t$ in practice. We therefore put greater emphasis on the expressivity of RWKV-7, and include time-varying $a_t$.

## D   Expressivity of RWKV-7

Transformers aggregate history through commutative sums, which limits their ability to solve sequential state tracking problems where the precise order of actions matters for the result (formally, these problems are outside $\mathsf{TC}^0$ under standard conjectures). A key example of state tracking

is computing the state of a chess board given a sequence of moves, each specified in notation like 'e2e4', indicating the source and target square. Transformers, S4 and Mamba cannot solve this chess state tracking problem beyond short input lengths if no extra 'thinking' tokens are given(Merrill et al., 2024; Merrill & Sabharwal, 2024). In contrast, our result shows that RWKV-7 can solve the chess state tracking task for any sequence length as long as the model has at least four layers with sufficient width, and a head size that is linear with respect to the number of board positions.

We show that the RWKV-7 architecture can express $\mathsf{NC}^1$-complete state tracking problems that cannot be expressed by transformers or other recurrent architectures such as S4 and Mamba, under standard complexity conjectures. We first show a particular $\mathsf{NC}^1$-complete problem that can be expressed by RWKV-7 in Section D and then generalize the argument to show that any regular language can be recognized by RWKV-7 in Section D.2. As regular language recognition can be understood to formalize finite state tracking problems, this suggests an expressivity advantage of RWKV-7 on state-tracking problems.

### D.1 Warmup: Expressivity Beyond $\mathsf{TC}^0$

Recall that the RWKV-7 wkv state is updated, at each token, by multiplication with $A_t = \text{diag}(w_t) - c\hat{\kappa}_t^T(a_t \odot \hat{\kappa}_t)$, where $c = 1$. In the following, we will consider $c = 2$, and show that this yields expressivity beyond $\mathsf{TC}^0$ (unless $\mathsf{TC}^0 = \mathsf{NC}^1$). $\mathsf{TC}^0$ is the complexity class which includes transformers as well as all non-gated or diagonal SSMs (Merrill et al., 2024; Barrington, 1989).

**Proof of Theorem 1**

*Proof.* RWKV-7 can, by Lemma 2, solve the problem of tracking swaps on five elements. This problem is $\mathsf{NC}^1$-complete under $\mathsf{AC}^0$ reductions (Merrill et al., 2024). $\qquad\square$

**Lemma 1.** *The RWKV-7 transition matrix can represent an arbitrary swap matrix, where a swap matrix is an identity matrix with two of its rows swapped.*

*Proof.* Given indices $x$ and $y$, let

$$w_t = 1, \quad c = 2, \quad \hat{\kappa}_t = (e_x - e_y)/\sqrt{2}, \text{ and } a_t = 1.$$

Here $e_i$ denotes the vector with 1 at position $i$ and 0 elsewhere.

Then the transition matrix becomes $A_t = I - e_x^T e_x - e_y^T e_y + e_x^T e_y + e_y^T e_x$, which is the permutation matrix that swaps indices $x$ and $y$. $\qquad\square$

**Lemma 2** (RWKV-7 can track swaps on 5 elements)**.** *Let a sequence of swaps on 5 elements be encoded in the format*

$$\#[x_1 \leftrightarrow y_1][x_2 \leftrightarrow y_n]\ldots,$$

*where # is a special beginning-of-sequence token, and $[x_i \leftrightarrow y_i]$ is a token that denotes a swap between elements $x_i$ and $y_i$. Then there exists a one-layer RWKV-7 model which outputs 1 if the sequence of swaps encode the identity permutation, and outputs 0 otherwise.*

*Proof.* Let the RWKV-7 model have 5 wkv heads of head dimension 5. The embedding weights and "weight preparation" part of the time mixing layer are set such that the following two properties hold:

Firstly, when the model sees the special beginning-of-sequence token, the $i$th wkv head receives

$$w = 1, \; c = 2, \; \hat{\kappa} = e_i, \; a = 0, \text{ and } v = \tilde{k} = e_i.$$

Here $e_i$ denotes the vector with 1 at position $i$ and 0 elsewhere. This sets the state of the $i$th wkv head to $\mathbf{wkv} = e_i^T e_i$, which represents state $i$.

Secondly, when token represents a swap between tokens $1 \leq x < y \leq 5$. In this case, using Lemma 1, all wkv heads receive

$$w = 1, \; c = 2, \; \hat{\kappa} = (e_x - e_y)/\sqrt{2}, \; a = 1, \text{ and } v = \tilde{k} = 0.$$

This changes the state to $y$ if it was $x$, or $x$ if it was $y$, or keeps it unchanged otherwise.

To calculate the output, the $i$th wkv head checks whether it represents state $i$, by applying receptance $r = e_i$. Finally, the MLP layer combines these outputs to check if all 5 heads agree with the identity permutation, and the output head outputs 1 if this is true, and 0 otherwise. $\qquad\square$

## D.2 Main Result: RWKV-7 Can Recognize Any Regular Language

Moreover, we are able to demonstrate that RWKV-7 has the capability to recognize any regular language. The regular languages are precisely those which can be recognized by a deterministic finite automaton (DFA). Therefore, it is sufficient to show that RWKV-7 can simulate any DFA. We define DFAs in the usual way:

**Definition 1.** *Classically, a DFA is a tuple $\mathscr{A} = (Q, \Sigma, \delta, q_0, F)$ where $Q$ is a finite set of states, $\Sigma$ is a finite vocabulary of tokens, $\delta_\sigma : Q \to Q$ is a transition function for each token $\sigma \in \Sigma$, $q_0 \in Q$ is the initial state, and $F \subseteq Q$ is a set of accepting states.*

*Equivalently, the DFA's computation on $w \in \Sigma^*$ can be represented by matrix computations. Each $\delta_\sigma$ can be represented by a boolean matrix $M_\sigma \in \{0, 1\}^{|Q| \times |Q|}$, where $M_w(i, j) = 1$ iff $\delta_w(q_j) = q_i$. The initial state $q_0$ can be represented as a one-hot vector $\alpha \in \{0, 1\}^{|Q|}$, and the set of accepting states $F$ can be represented as a multi-hot vector $\omega \in \{0, 1\}^{|Q|}$.*

*For a given string $w_1 \cdots w_T$, the DFA computes*

$$\alpha \cdot M_{w_1} \cdots M_{w_T} \cdot \omega^\top \tag{25}$$

*We say that $w \in L$ if and only if this expression evaluates to 1.*

Having defined regular languages, we are in a position to show our main result:

**Proof of Theorem 2**

*Proof.* We demonstrate that the RWKV-7 architecture can recognize strings in an arbitrary regular language $L$. Consider a string $w \in \Sigma^*$ and its membership in $L$. There exists a DFA (Definition 1) which recognizes $L$ by evaluating (25). To prove that RWKV-7 can recognize any regular language, it is therefore sufficient to construct a RWKV-7 that evaluates (25).

**Construction overview.** A standard way to recognize a regular language would be to construct the transition matrices for each token and then multiply them to compute the final state. However, this does not work for RWKV-7 because because an arbitrary DFA transition can have rank $n = |Q|$ (i.e., the number of states), whereas a wkv head can only implement a simple elementary transition matrix per input token. A natural idea is to factor each DFA transition into *n elementary transition matrices* (Lemma 3), each of which can be directly implemented by wkv heads (Lemma 4). However, expanding DFA transitions in this way gives more elementary transition matrices than tokens, which means it cannot be directly implemented by RWKV-7.

Fortunately, a simple modification of this idea will allow us to implement regular language recognition in RWKV-7. Rather than expanding each transition matrix to $n$ elementary matrices, we use the first three layers to convert blocks of $n$ DFA transitions to products of $n$ elementary matrices with the same product. The final layer then multiplies these elementary matrices to obtain the final state.

**Details: information routing via residual stream and Layernorm.** The output of each layer is stored in the residual stream of the architecture, in independent subspaces, which makes them all available to deeper layers. Since the outputs come from a finite set, they may be represented by one-hot encoding.

The input to each time mixing block therefore contains the outputs of all previous layers. First, a Layernorm is applied. Note that one-hot encodings have constant norm, which ensures that the Layernorm preserves the encoded information.

Next is the "weight preparation" part of the time mixing block, which takes input $x_t$ encoding the output from previous layers, and constructs $r_t, w_t, \tilde{k}_t, v_t, \hat{\kappa}_t$ and $a_t$ for the wkv heads. The weight preparation is sufficiently expressive to allow each of the 6 output variables to be an arbitrary

linear transform of $x_t$. This allows selecting $r_t, w_t, \tilde{k}_t, v_t, \hat{\kappa}_t$ and $a_t$ based on arbitrary outputs from previous layers. Therefore, the wkv heads are the main part of the construction.

**WKV head construction.** For any $1 \le t \le T$, the current position $t$ can be split into $t = ln + \hat{t}$, for integers $0 \le l$ and $1 \le \hat{t} \le n$. We view the input sequence as blocks of length $n$, indexed by $l$.

First, we describe $l \ge 1$. Consider the product of DFA transitions $\tilde{M}_l = M_{w_{(l-1)n+1}} \dots M_{w_{(l-1)n+n}}$. This product is also a DFA transition matrix (i.e., it has a single 1 per column). Hence, Lemma 3 allows us to factor $\tilde{M}_l = G_{l,1} G_{l,2} \dots G_{l,n}$, where $G_{l,1}, \dots, G_{l,n}$ are elementary transition matrices. Fix one such factorization for each possible DFA transition matrix. At position $t = ln + \hat{t}$, the wkv state in the fourth (final) layer is right-multiplied by the elementary transition matrix $G_{l,\hat{t}}$. Since $G_{l,\hat{t}}$ is uniquely defined by the last $2n$ tokens and position modulo $2n$, it can be computed in the third layer by Lemma 7, and fed to the fourth layer.

The first block $l = 0$ is handled as a special case. Again, we consider the fourth layer. At the first token, set $v_1 = e_1$ and $\tilde{k}_1 = \alpha$. All subsequent transitions $t \ge 2$ set $v_t = \tilde{k}_t = \mathbf{0}$, and implement identity transition matrices from Lemma 4. Then $\mathbf{wkv}_t = e_1^T \alpha$ for $1 \le t \le n$.

With this construction, for all $1 \le t \le T$, the fourth layer's wkv state becomes

$$\mathbf{wkv}_t = e_1^T \hat{\alpha}_t, \text{ where } \hat{\alpha}_t = \alpha M_{w_1} M_{w_2} \dots M_{w_{(l-1)n}} G_{l,1} G_{l,2} \dots G_{l,\hat{t}},$$

where as usual, empty products (such as for $l \le 1$) evaluate to identity matrices.

Duplicate this construction into $n$ wkv heads, where the $i$th applies receptance $e_i$. In combination, the wkv heads read out the whole vector $\hat{\alpha}_t$.

**Final tokens.** Consider $t = T$. To match the DFA evaluation formula from (25), $\hat{\alpha}_T$ can be multiplied by

$$\hat{\omega}_T^T = G_{l,\hat{t}+1} \dots G_{l,n} M_{w_{ln+1}} M_{w_{ln+2}} \dots M_{w_{ln+\hat{t}}} \omega^T.$$

Fortunately, this expression is a fixed function of the current position modulo $2n$ and the last $2n$ tokens, and can hence be found by a lookup in the third layer by Lemma 7. The final MLP on the final token can thus output the scalar product $\hat{\alpha}_T \hat{\omega}_T^T$, which is equal to (25), thus completing the construction. $\qquad \square$

Our construction uses MLPs to implement lookup tables with sizes on the order of $|\Sigma|^{2n}$, which may require MLP layers that are exponentially wide in the number of states of the original DFA.

### D.3 Lemmas for Theorem 2

The proof for Theorem 2 requires many different lemmas, which are stated and proved in the following.

A single RWKV-7 wkv state transition cannot directly implement an arbitrary DFA transition. However, DFA transition matrices can be decomposed into a product of *elementary transition matrices* that can be directly simulated by wkv state transitions (cf. Lemma: 1):

**Lemma 3.** *Let $M$ be a DFA transition matrix. I.e., $M$ has shape $n \times n$, and contains a single 1 in each column. Then $M$ can be factored into a product of $n$ elementary transition matrices $G_1, \dots, G_n$. Specifically,*

$$M = G_1 G_2 \dots G_n,$$

*where each of $G_1, \dots, G_n$ has one of the following forms:*

1. *Identity matrix.*
2. *Swap matrix $x \leftrightarrow y$; an identity matrix with rows $x$ and $y$ swapped.*
3. *Copy matrix $x \rightarrow y$; an identity matrix with column $y$ replaced by a copy of column $x$.*

*Proof.* We will greedily build $M$ from the identity matrix by right-multiplying elementary transition matrices. Right-multiplying by the identity matrix does nothing, right-multiplying with a swap matrix $x \leftrightarrow y$ swaps columns $x$ and $y$, and right-multiplying with a copy matrix $x \rightarrow y$ replaces column $x$ by a copy of column $y$.

We use $X$ to denote the current partial product of elementary transition matrices. Initially, $X$ is the identity matrix, and the goal is to apply $n$ transitions to make $X = M$. We proceed greedily in three stages.

1. Find a column $c$ of $X$ that differs from column $c$ of $M$, but which matches a different column $c'$ of $M$. If no such position exists, proceed to the next stage. Otherwise, right-multiply $X$ by a swap matrix that swaps columns $c$ and $c'$.
   Note that $M$ and $X$ differed in columns $c$ and $c'$ before the swap, but match in column $c$ after the swap.
2. Find a column $c$ where $M$ and $X$ differ. If no such column exists, proceed to the next stage. Otherwise, the previous stage has ensured that there exists a column $c'$ of $X$, necessarily different from $c$, which contains a column identical to column $c$ of $M$.
   Right-multiply by a copy matrix that replaces column $c$ of $X$ with column $c'$ of $X$. Note that $M$ and $X$ differed in column $c$ before the move, while they agree afterwards.
3. Right-multiply by identity matrices until $X$ is the product of $n$ elementary transition matrices.
   Initially, $M$ and $X$ differed in at most $n$ columns. Each subsequent right-multiplication by an elementary transition matrix reduced the number of differing columns by at least one. Thus, stages 1 and 2 required at most $n$ multiplications.

$\square$

Recall that the RWKV-7 wkv state is at token index $t$ updated by multiplication with

$$A_t = \mathrm{diag}(w_t) - c\hat{\kappa}_t^T(a_t \odot \hat{\kappa}_t),$$

with $c = 1$. To simplify the presentation of the core idea, we will instead present a construction with $c = 2$, and then show how to remove this assumption in Section D.3.

**Lemma 4.** *For any elementary transition matrix $G$ (in the sense of Lemma 3), there exist $n$-dimensional vectors $\hat{\kappa}$ and $\vec{a}$, where $\|\hat{\kappa}\| = 1$, $\vec{a}$ has elements in $\{0, 1\}$, and*

$$G = \mathrm{diag}(w) - c\hat{\kappa}^T(\vec{a} \odot \hat{\kappa}),$$

*where $c = 2$ and $w = \mathbf{1}$.*

*Proof.* We use $e_i$ to denote the $n$-dimensional vector with 1 at position $i$ and 0 elsewhere.

1. Identity matrix: Select any length one vector $\hat{\kappa}$, for example $\hat{\kappa} = e_1$, and $\vec{a} = \mathbf{0}$.
2. Swap matrix; an identity matrix with rows $x$ and $y$ swapped: Select $\hat{\kappa} = (e_x - e_y)/\sqrt{2}$ and $\vec{a} = \mathbf{1}$.
3. Copy matrix; an identity matrix with column $y$ replaced by a copy of column $x$: Select $\hat{\kappa} = (e_x - e_y)/\sqrt{2}$ and $\vec{a} = e_x$.

$\square$

We now move on to the explicit construction of the first three layers.

**Lemma 5.** *There is a 1-layer RWKV-7 which outputs whether the current position is first, and whether the current position is even or odd.*

*Proof.* RWKV-7 performs a token-shift operation before the wkv heads are reached. This token shift takes as input the last token, and can therefore detect whether a previous token exists.

The first layer's wkv state can track position parity by selecting $\tilde{k}_1 = v_1 = e_1$ for the first position $t = 1$, and subsequently letting $c = 2$, $w_t = a_t = \mathbf{1}$, $\hat{\kappa}_t = e_1$ and $\tilde{k}_t = v_t = \mathbf{0}$ for $t \geq 2$. This leads to $\mathbf{wkv}_1 = e_1^T e_1$ and subsequently $\mathbf{wkv}_t = \mathbf{wkv}_{t-1}(I - 2e_1^T e_1)$ for $t \geq 2$. Then $\mathbf{wkv}_t = e_1^T e_1$ for odd $t$ and $\mathbf{wkv}_t = -e_1^T e_1$ for even $t$.

Receptance $r = e_1$ can then be used to read out the sign of the wkv state, which encodes the current position's parity.

The subsequent MLP can transform these two boolean outputs to an arbitrary format. $\square$

**Lemma 6.** *For any positive integer n, there is a 2-layer RWKV-7 that outputs the position modulo* $2n$.

*Proof.* We use Lemma 5 for the first layer, which tells the second layer whether the current position is first, and the parity of the current position.

At the first position, set $\tilde{k}_1 = v_1 = e_1$, such that $\mathbf{wkv}_1 = e_1^T e_1$. For all subsequent positions $t \geq 2$, set $c = 2$, $w_t = a_t = \mathbf{1}$ and $\tilde{k}_t = v_t = \mathbf{0}$. Furthermore, set $\hat{\kappa}_t = e_1$ for even $t$ and $\hat{\kappa}_t = \cos(\pi/n)e_1 + \sin(\pi/n)e_2$ at odd $t$. Then $\mathbf{wkv}_T = e_1^T e_1(I - 2\hat{\kappa}_2^T \hat{\kappa}_2)\ldots(I - 2\hat{\kappa}_T^T \hat{\kappa}_T)$. Note that for even $t \geq 2$, the matrix $(I - 2\hat{\kappa}_t^T \hat{\kappa}_t)(I - 2\hat{\kappa}_{t+1}^T \hat{\kappa}_{t+1})$ rotates the first two coordinates by an angle $2\pi/n$. Thus,

$$\mathbf{wkv}_t = \begin{cases} e_1^T (\cos(\pi(t-1)/n)e_1 + \sin(\pi(t-1)/n)e_2), & \text{if } t \text{ is odd} \\ e_1^T (-\cos(\pi(t-2)/n)e_1 + \sin(\pi(t-2)/n)e_2), & \text{if } t \text{ is even} \end{cases}.$$

The wkv heads are immediately followed by group normalization, which discards information about magnitudes. We therefore use $2n$ wkv heads of the type above, where the $k$th head applies receptance $r = \cos(\pi k/n)e_1 + \sin(\pi k/n)e_2$. The signs of these readouts, along with the parity from the first layer, can then be combined in the subsequent MLP layer to deduce the current position modulo $2n$. □

**Lemma 7.** *Let* $\tilde{t} \equiv t$ *modulo* $2n$ *be the current position modulo* $2n$, *and let* $w_t, w_{t-1}, \ldots, w_{t-(2n-1)}$ *be the last* $2n$ *tokens. Define* $w_t = |\Sigma| + 1$ *for before-sequence tokens* $t \leq 0$. *Let* $\Xi[\tilde{t}, w_t, w_{t-1}, \ldots, w_{t-(2n-1)}]$ *be a lookup table that takes as key the current position modulo* $2n$ *and the* $2n$ *most recent tokens.*

*Given inputs* $\tilde{t}$ *and* $w_t, w_{t-1}, \ldots, w_{t-(2n-1)}$, *there is a layer of RWKV-7 that simulates* $\Xi$.

*Moreover, there is a 3-layer RWKV-7 that simulates* $\Xi$.

*Proof.* Recall that the wkv state is initialized to all zeros. We apply the wkv state update

$$\mathbf{wkv}_t = \mathbf{wkv}_{t-1}(I - e_{\tilde{t}}^T e_{\tilde{t}}) + e_{w_t}^T e_{\tilde{t}}.$$

This can be achieved by selecting $c \geq 1$, $w_t = \mathbf{1}$, $a_t = \frac{1}{c}\mathbf{1}$, $\tilde{k}_t = \hat{\kappa}_t = e_{\tilde{t}}$ and $v = e_{w_t}$.

In words, the state update replaces the $\tilde{t}$th column of the wkv state with $e_{w_t}$. Hence, the wkv state stores the last $2n$ tokens.

We make $n$ such wkv heads, where the $i$th wkv head applies receptance $r = e_i$. This reads out the full state, which contains the last $2n$ tokens. The state and $\tilde{t}$ are fed into the subsequent MLP layer, which performs the lookup into $\Xi$.

Note that by Lemma 6, the first two layers can compute current position modulo $2n$ and the $2n$ most recent tokens. Hence, a 3-layer RWKV can compute $\Xi$. □

**Removing the assumption** $c = 2$   Some of our constructions use $c = 2$, while the actual model uses $c = 1$. However, since the transition matrix is $A_t = \text{diag}(w_t) - c\hat{\kappa}_t^T(a_t \odot \hat{\kappa}_t)$, halving both $c$ and $w_t$ simply causes $A_t$ to be halved. This causes the wkv state to halve in magnitude at each token. However, since the wkv heads are immediately followed by group normalizations, the magnitude of the wkv state does not affect subsequent calculations. Additionally, since floating point numbers store a separate exponent, this rescaling only requires log-precision. The shrinking state could in principle be mismatched with the scales of $v_t$ and $\tilde{k}_t$, but our constructions always satisfy $v_t = \tilde{k}_t = \mathbf{0}$ whenever $c \neq 1$ is required.

# E   Additional Architectural and Training Details

**Training Method and Model Data Amounts**   The RWKV-7 Pile models all use the GPT-NeoX-20B tokenizer (Black et al., 2022), and were all trained from scratch on the Pile dataset, which has 332 billion tokens.

All RWKV World dataset models use the RWKV World Tokenizer. Due to compute budget constraints, the Goose World 3 0.1B and 0.4B models were trained from pre-existing RWKV-5 World v1

and v2 checkpoints, and the Goose World 3 1.5B and 2.9B models were trained from pre-existing RWKV-6 World v2.1 checkpoints. These checkpoints' parameters were then converted to the RWKV-7 format via a process described below. Once in the new format, the models are trained on either the additional full 3.1 trillion tokens of the World v3 corpus, or an equally weighted sub-sampling of it. Under this methodology, some documents were seen two or even three times.

The World v1, v2, v2.1, and v3 corpora contain 0.6, 1.1, 1.4, and 3.1 trillion tokens, respectively. The amounts of training in each stage at with each successive model architecture and corpus are shown in Table 10.

| Model | World v1 | World v2 | World v2.1 | World v3 | Total |
|---|---|---|---|---|---|
| RWKV7-World3-0.1B | 0.6 (RWKV-5) | | | 1.0 (RWKV-7) | 1.6 |
| RWKV7-World3-0.4B | | 1.1 (RWKV-5) | | 2.0 (RWKV-7) | 3.1 |
| RWKV7-World3-1.5B | | 1.1 (RWKV-6) | 1.4 (RWKV-6) | 3.1 (RWKV-7) | 5.6 |
| RWKV7-World3-2.9B | | 1.1 (RWKV-6) | 1.4 (RWKV-6) | 3.1 (RWKV-7) | 5.6 |

Table 10: Total trillions of tokens trained for all RWKV-7 World 3 models

Our model format conversion process involves removing the token-shift low-rank MLPs, rescaling by half the embeddings, wkv receptance, wkv output matrix weights, and Layernorm and Groupnorm bias values. Layernorm and Groupnorm weights are clamped above zero and square rooted. We widen the FFN MLP from 3.5x (in RWKV-6) to 4x and add new small ($1 \times 10^{-3}$) uniform initalizations in the new regions, removing the RWKV-6 FFN receptance weights. We widen the time decay Low-rank MLP and add new small ($1 \times 10^{-4}$) uniform initializations in the new regions. We replace the gate weights with a LoRA obtained through singular value decomposition and rescaling by half.

**Architecture Diagram** We provide a comprehensive architecture diagram (Figure 5) in order to help readers thoroughly understand our architecture design.

**Parameters and Dimensions** Throughout this section, we denote by $D$ the model dimension, $L$ the number of layers, $h = D/D_h$ the number of heads, and $V$ the vocabulary size. All models are trained with head size $D_h = 64$, i.e., each time-mixing has $h = D/64$ heads with dimension $64 \times 64$.

Pile models are trained with $V = 50304$ with the GPT-NeoX 20B tokenizer. World models are trained with $V = 65536$ with RWKV World tokenizer.

| Model Name | $L$ | $D$ | State Size (WKV + Shift) | Parameters |
|---|---|---|---|---|
| RWKV7-World3-0.1B | 12 | 768 | $589\,824 + 18\,432$ | $191\,034\,624$ |
| RWKV7-World3-0.4B | 24 | 1024 | $1\,572\,864 + 49\,152$ | $450\,767\,872$ |
| RWKV7-World3-1.5B | 24 | 2048 | $3\,145\,728 + 98\,304$ | $1\,527\,404\,544$ |
| RWKV7-World3-2.9B | 32 | 2560 | $5\,242\,880 + 163\,840$ | $2\,947\,735\,040$ |

Table 11: Released Goose models with parameters and state size.

RWKV-7 uses four low-rank MLPs for decay $w$, value residual $v$, in-context learning rate $a$ and gate $g$ respectively. The intermediate dimensions are listed in Table 12. These values are based on our mere speculation of how much information can be passed through.

The number of parameters for all RWKV-7 models can be computed by the formula:

$$\#(\text{Params}) = 2DV + 4D + LD\left(12D + 2\left(d_w + d_a + d_v + d_g\right) + 19\right) - (2Dd_v + D). \tag{26}$$

Where:

- The weights of the embeddings and head, and the Layernorms beside them, yield $2DV + 4D$ parameters;

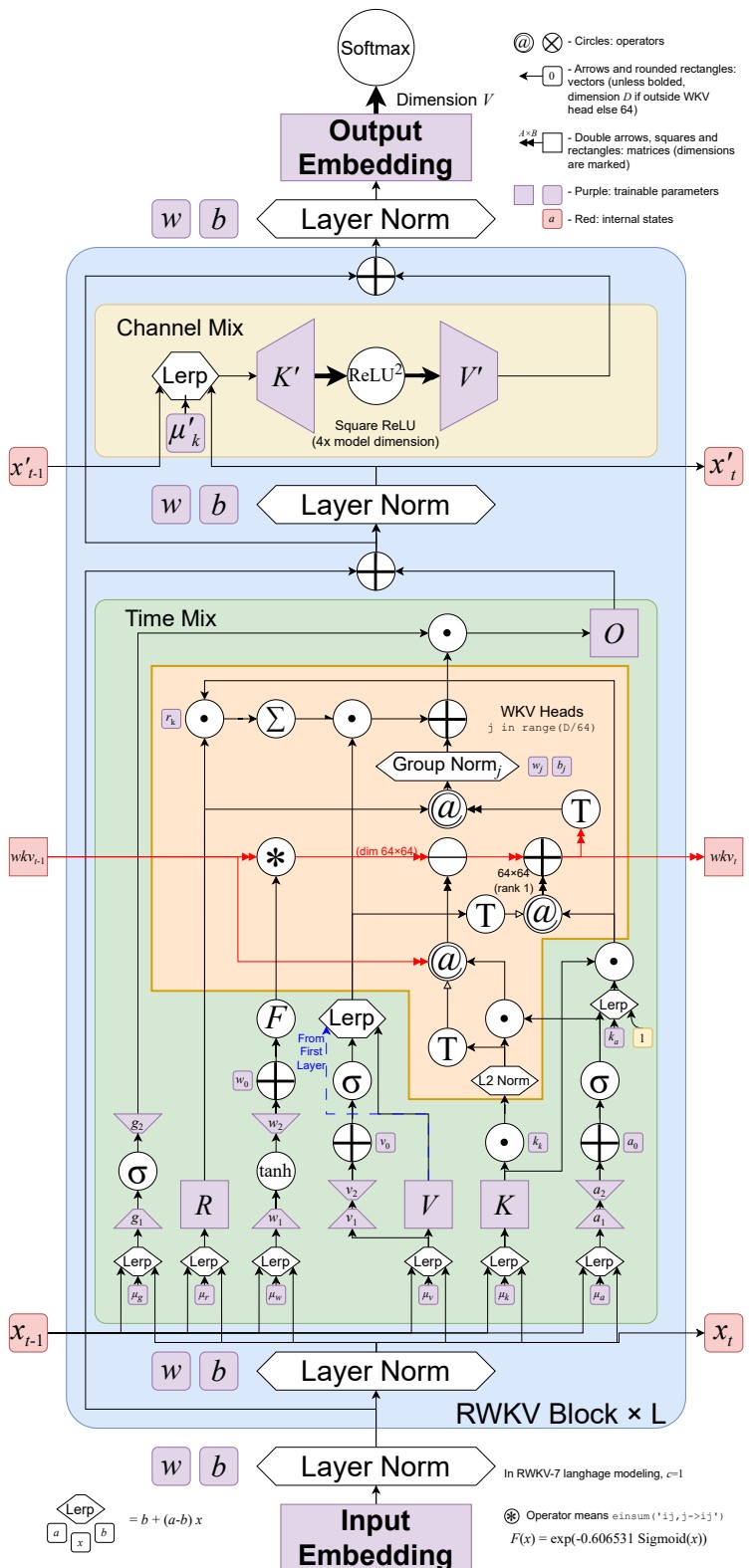

Figure 5: The architecture of RWKV-7, drawn in detail.

| Dimension ($D$) | $d_w$ | $d_a$ | $d_v$ | $d_g$ |
|---|---|---|---|---|
| 768 | 64 | 64 | 32 | 128 |
| 1024 | 64 | 64 | 32 | 128 |
| 2048 | 96 | 96 | 64 | 256 |
| 2560 | 96 | 96 | 64 | 320 |
| 4096 | 128 | 128 | 96 | 480 |
| 6144 | 128 | 128 | 96 | 640 |

Table 12: Suggested intermediate dimensions for the low-rank MLPs for RWKV-7 models

- The weights of each layer yield $D\left(12D + 2\left(d_w + d_a + d_v + d_g\right) + 19\right)$ parameters, except for the first layer;
- The low-rank MLP for the value residual is not present in the first layer, subtracting $(2Dd_v + D)$ parameters.

**Parameter Initializations**  Proper parameter initialization is crucial for ensuring training stability and achieving optimal performance for language models. RWKV-7 employs a carefully designed initialization strategy tailored to its architecture. The detailed initialization scheme is beyond the scope here but can be found in the official code repository. We emphasize that using the recommended initialization is essential for replicating the results in this paper. Deviations from the prescribed initialization may lead to performance degradation.

**Dataset Loading**  The dataset used for pretraining consists of $3\,119\,194\,079\,123$ tokens stored on disk, which are memory-mapped using the `mmap` mechanism. To ensure a diverse and pseudo-random sampling of training sequences, we employ a custom data loading strategy based on a mathematical function with desirable properties. Specifically, we utilize a pseudo-random number generator defined by the function $f(x) = ax^3$ over the finite field $\mathbb{Z}/p\mathbb{Z}$, where $p$ is a prime number of the form $3n + 2$. This function is chosen because it is a bijection (full map) in $\mathbb{Z}/p\mathbb{Z}$, ensuring that all possible indices are eventually accessed exactly once within one epoch.

For pretraining with a sequence length of 4096, the relative address of the $k$-th sample is determined as:

$$\text{start\_address} = 4096 \cdot (ak^3 \mod p), \quad \text{end\_address} = \text{start\_address} + 4097.$$

Here, $p$ is chosen as the largest prime of the form $3n + 2$ smaller than $\lfloor \text{dataset\_size}/4096 \rfloor$, yielding $p = 761\,521\,949$. The parameter $a$ is set to an integer close to $0.618p$ that ensures good mixing properties of the generated addresses.

This approach guarantees both simple calculation and uniform access to the dataset while maintaining pseudo-randomness. By leveraging the properties of modular arithmetic and cubic mappings, we achieve a balance between computational efficiency and data diversity during pretraining.

**Training Details**  All RWKV-7 models were trained under `bfloat16` format on nodes of 8× Nvidia H800. The AdamW optimizer was configured with $\beta_1 = 0.9$, $\beta_2 = 0.99$, $\epsilon = 1 \times 10^{-18}$, and a weight decay of 0.1 applied exclusively to linear layers and embedding weights. The choice of such a small $\epsilon$ value is motivated by the theory proposed by Molybog et al. (2023), which suggests that reducing $\epsilon$ can help stabilize training in large-scale models by ensuring that intermediate layers remain in a regime of active updates, thus mitigating sudden loss spikes and promoting smoother convergence.

The context length for pretraining was 4096 tokens. The base decay rate $w_0$ parameters are placed into a special 2x learning rate multiplier grouping.

Besides the traditional cosine learning rate decay schedule, we used a phased dynamic batch size scaling strategy inspired by the concept of critical batch size proposed by McCandlish et al. (2018) and similar to the approaches in Smith et al. (2018). Our strategy involves progressively increasing the batch size during training, accompanied by corresponding adjustments to the learning rate.

| Model | Phase | Nodes | Batch size | Proposed Initial LR | Final Loss |
|-------|-------|-------|------------|---------------------|-----------|
| RWKV7-World3-0.1B | 1 | 1 | $240 \times 4096$ | $6 \times 10^{-4}$ | 2.5290 |
| RWKV7-World3-0.4B | 1 | 1 | $240 \times 4096$ | $5 \times 10^{-4}$ | |
| | 2 | 2 | $480 \times 4096$ | $6 \times 10^{-4}$ | 2.2580 |
| RWKV7-World3-1.5B | 1 | 3 | $480 \times 4096$ | $4 \times 10^{-4}$ | |
| | 2 | 4 | $672 \times 4096$ | $4.5 \times 10^{-4}$ | |
| | 3 | 6 | $1152 \times 4096$ | $6.1 \times 10^{-4}$ | 1.9970 |
| RWKV7-World3-2.9B | 1 | 4 | $640 \times 4096$ | $4 \times 10^{-4}$ | |
| | 2 | 6 | $1008 \times 4096$ | $5 \times 10^{-4}$ | |
| | 3 | 7 | $1120 \times 4096$ | $5.4 \times 10^{-4}$ | |
| | 4 | 12 | $2016 \times 4096$ | $8 \times 10^{-4}$ | 1.8745 |

Table 13: Training schedules and batch sizes

The detailed training schedules for different model sizes are listed in Table 13.

The learning rate undergoes a cosine decay schedule from the proposed initial learning rate at the beginning of the entire training run to the expected final learning rate of $1 \times 10^{-5}$ at the end of the entire run, but the implied initial rate varies across phases.

This approach not only enhances training efficiency but also utilizes GPU resources economically. After smaller models complete their training, additional GPU resources become available for the later stages of training larger models. This cascading resource allocation ensures that computational power is dynamically reallocated, maximizing hardware utilization and reducing idle time.

We observe extremely stable training without any loss spikes in all four runs, indicating that the likelihood of encountering such spikes during the training of a very large RWKV-7 model is minimal.

See Figure 6 for the resulting learning rates and observed loss curves.

Despite the general stability of our loss curves, we did sometimes observe NaN loss across a single training step, which we theorize may be due to our use of such an extremely low AdamW $\epsilon$. When this occurs, we rewind the training to the prior checkpoint, clear optimizer states, and continue from that point.

## F    Additional Architecture Discussion

The following is a general overview of the RWKV-7 architecture and selected rationale for its design choices:

The RWKV-7 Time Mixer begins with a feature that has been in RWKV since its inception: token shift. Token shift is a variety of 1D short convolution that is intended to allow the model to create induction heads within a single layer. At its core, token shift is just a linear interpolation between the current and prior tokens on a per channel basis, where the amount per channel is a learned parameter. Many other modern models (e.g. Mamba) have begun including short convolutions before attention replacements (See also Dao AI Lab (2023)). RWKV-6 introduced an advanced new form of token shift that was data dependent. While we found that this was beneficial in terms of loss decrease per step, we made the judgement call that the improvement in training and inference efficiency was not worthwhile. Therefore, RWKV-7 includes only the simple token shift found in RWKV-4 and RWKV-5.

RWKV-7 time mixing follows the overall form of delta rule, applying an SGD-like mechanism to train the state at test time, such that when presented with a key it will respond with an appropriately matching value, like those it has been shown previously. This can be conceptualized as a simple order of operations: 1) decay the state 2) remove part of the old value located at the current key 3) add part of the new value into the current key. Then, a query (we call it receptance) is applied to

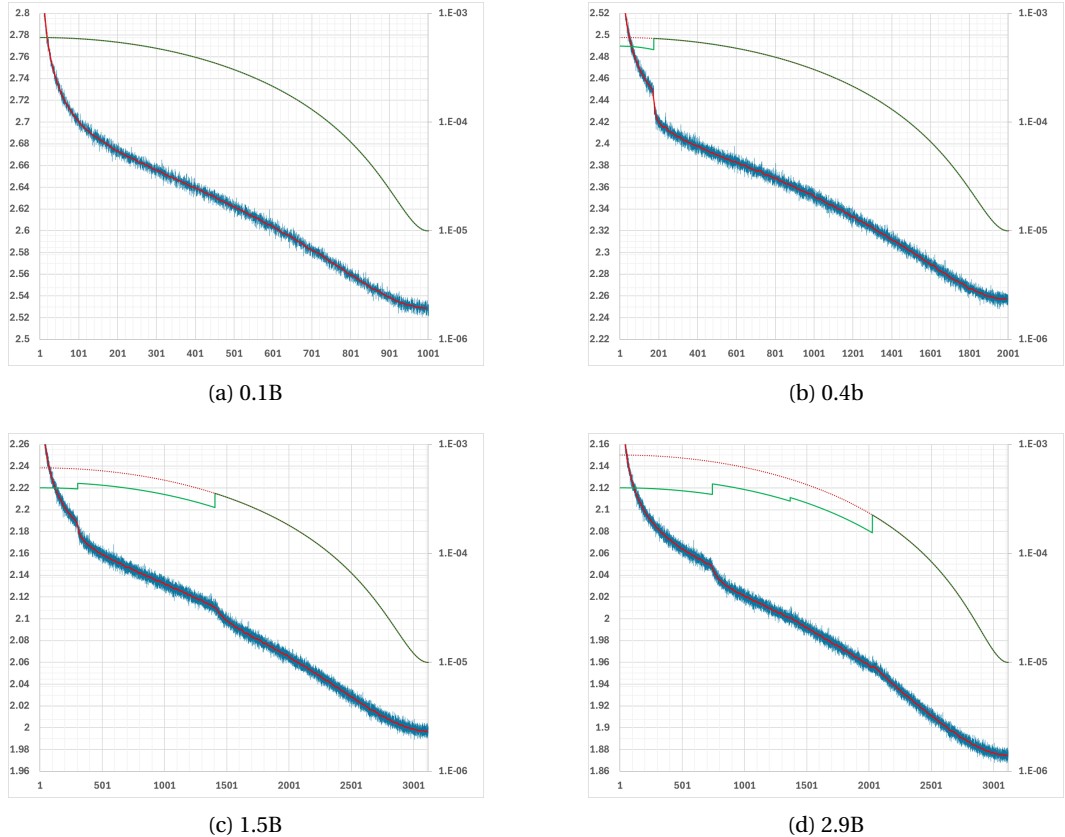

(a) 0.1B
(b) 0.4b
(c) 1.5B
(d) 2.9B

Figure 6: Training loss curve for RWKV-7 World models. Blue line: loss; Red line: smoothed loss; Green line: actual learning rate; Dotted red line: learning rate schedule.

the state in order to request the value located at a specific key. After that, the remaining operations normalize the returned values to keep numerical sizing consistent, apply gating, and recombine the heads to obtain the output.

In SGD terms, the decay found in models like RWKV-7 and Gated DeltaNet can be thought of as similar to weight decay. RWKV models since RWKV-4 have employed some form of vector-valued decay on the state in place of positional encoding. We continue this tradition with RWKV-7. While vector valued decay is harder to compute efficiently than its scalar valued equivalent, it brings a significant improvement in loss per step. Many other models, like Mamba-2, make the choice to maximize training efficiency and simplify their kernels by using scalar decay, purposely trading quality for speed.

We are forced to limit the maximum decay that can occur in a given time-step both in order to maintain training stability and to assist the creation of fast but numerically stable kernel implementations. We have many varieties of kernel available today, and are still working on new designs with enhanced efficiency and accuracy. Please see Parallelizing Linear Transformers with the Delta Rule over Sequence Length (Yang et al., 2024c) and its accompanying blog post series for insightful details on many components of such algorithms. We hope to be able to reduce the decay limit further in future revisions.

This lower bound on the decay multiplier is expressed by the $\exp(-e^{-0.5}\sigma(d_t))$ formula for decay. It is a rearrangement of an original source formula $\exp(-\exp(-0.5 - \text{softplus}(d_t)))$. The outer $\exp(-\exp(x))$ in the original is nearly a flipped version of sigmoid, but with a better gradient. The $-0.5 - \text{softplus}(d_t)$ is a way of limiting the inputs to this sigmoid-like function to be less than -0.5, which results in the final decay being greater than 0.545. This means that decay can remove at most 45.5% of the pre-existing values in the wkv state from one timestep to the next. More can be then removed by the delta rule mechanism.

The in-context learning rate is usually equivalent to the learning rate in SGD. Ours is a bit less restrictive than the traditional delta rule or SGD would allow. We also make this data dependent and extend it to be vector-valued instead of being merely a scalar. This is allows each key channel to have its own learning rate, dependent on the current token. Note that in TTT, Gated DeltaNet, and Titans although the in-context learning rates are data dependent, they are only scalar valued.

Some of the design of RWKV-7 with regard to the in-context learning rate is theoretically motivated but may not be apparent from the equations. RWKV-6c, a later v6 sub-version with no trained models but which worked well experimentally, kept its state somewhat normalized using a new design. As mentioned early in this paper and consistent with the observations in Yang et al. (2024b); Chen et al. (2024a), there is a fundamental issue with linear attention numerically growing in an unbounded fashion, and the base RWKV-6 revision has this problem. RWKV-6c, however, defeats this issue by ensuring that the amount of key added to the state is never more than the amount removed by decay. It accomplishes this by pre-multiplying the key by one minus the decay before adding it into the wkv state.

Early versions of RWKV-7 attempted to use similar mathematical formulations to keep everything normalized. But experimentally we found that the model both did best and was most efficient when we allowed it enough mathematical leeway to make these decisions on its own, rather than enforcing them. So instead of pre-multiplying the key, we give the replacement key latitude in its learning rate via the replacement rate booster.

Similarly, the removal key is decoupled from the replacement key. Note that the removal key is normalized. This is important in delta rule variations, because otherwise the outer product of removal key with itself would cause unwanted changes in the amount removed due to the implicit squaring of its length during that process.

Another point that may not be clear upon first examination is the importance of RWKV-7 being outside of the $\mathsf{TC}^0$ complexity class in which transformers, linear attention, and many varieties of SSMs lie. Consider a single layer of the Key Value Cache of a transformer. It is appended to upon each new token processed, and can never be changed. RWKV-7, however, can and does edit its state at each subsequent token. This can include simple operations like replacement, which can be viewed as functioning similarly to a transformer losing a KV Cache entry via some sparsity mechanism e.g. a sliding window. But it can also include complex operations like swapping two entries in the state; operations a transformer cannot do within its immutable KV Cache. These kinds of operations can be used to enact computations on the state itself, which lends greater computational abilities to RWKV-7 when processing a fixed set of inputs. You might think of the RWKV-7 state as being like an internal scratchpad.

There are easy problems that simply cannot be solved by transformers on fixed inputs because they lack this ability. One example is that if you give a transformer an ordered set of items and a list of which ones swap places, it will not be able to tell you which item ends up in which position at the end. Both architectures gain even more power when they are allowed extra outputs (e.g. chain of thought), becoming Turing complete. But this requires costly additional inference-time computation, whereas the RWKV-7 state does not.

A possible way to interpret the designation of RWKV-7 is training a model to learn (how to train an internal model). A WKV head can be viewed as a linear transformation that takes the input (receptance) $r$ and outputs $o$. Every WKV head can be regarded as a linear model that can update its weights as the context progresses. The entries of WKV states are exactly the parameters of these models.

After receptance is applied to the WKV state, we normalize the result on a per head basis. This has become common across many linear attention and post-linear-attention architectures. It is a way of ensuring that change in the numerical size of the state over time does not impact the model's ability to use the state. The original formulations of RWKV-4 used a denominator in place of normalization to achieve a similar effect, but it is costly, harder to code, and uses more memory.

The readout part of RWKV-7 differ from RWKV-6 by the addition of the per-head $\left(r_t(\rho \odot \tilde{k}_t)^T\right)$ scalar gating of $v_t$. The trainable parameter $\rho$ resembles the design of "time-first" $u$ term existing from RWKV-4 to RWKV-6, under the belief that the information of the current token deserves special treatment. The "time-first" term was fused inside the WKV kernel in previous architectures

of RWKV, but we decide to extract this term out of the kernel to simplify the implementation of RWKV-7.

## G    Pseudocode For RWKV-7

```python
import math
import torch as th
import torch.nn.functional as F

def rwkv_timemix(params, x, vprime_0, shift_state, wkv_state, layer_id):
    B, T, C = x.shape       # batch_size, sequence_length, d_model
    N = wkv_state.shape[-1] # head size
    H = C // N              # head count

    # weight preparation
    x_shifted   = th.cat([shift_state, x[:, :-1, :]], dim=1)
    shift_state = x[:, -1:, :]

    x_receptance = th.lerp(x, x_shifted, params.mu_r)
    x_decay      = th.lerp(x, x_shifted, params.mu_d)
    x_key        = th.lerp(x, x_shifted, params.mu_k)
    x_value      = th.lerp(x, x_shifted, params.mu_v)
    x_iclr       = th.lerp(x, x_shifted, params.mu_a)
    x_gate       = th.lerp(x, x_shifted, params.mu_g)

    r      = x_receptance @ params.W_receptance # BTC
    d      = params.decay_lora(x_decay)         # BTC
    k      = x_key @ params.W_key               # BTC
    vprime = x_value @ params.W_value           # BTC
    gate   = params.gate_lora(x_gate)           # BTC
    iclr   = params.iclr_lora(x_iclr).sigmoid() # BTC

    # 1st layer: return value to use in later layers, no interpolation
    if layer_id == 0:
        v = vprime_0 = vprime
    else:
        value_residual_gate = th.sigmoid(params.nu_lora(x_value))
        v  = th.lerp(vprime, vprime_0, value_residual_gate)

    decay = th.exp(-math.exp(-0.5) * d.to(th.float).sigmoid())
    removal_k = k * params.removal_key_multiplier
    removal_k = F.normalize(removal_k.view(B,T,H,-1), dim=-1).view(B,T,C)
    replacement_k = th.lerp(k, k * iclr, params.iclr_mix_amt)

    # recurrence relation
    out = th.empty_like(x).view(B,T,C)
    for t in range(T):
        # single step of wkv state transition
        decay_t       = decay[:, t].view(B, H, N, 1)
        iclr_t        = iclr[:, t].view(B, H, N, 1)
        removal_k_t   = removal_k[:, t].view(B, H, N, 1)
        replacement_k_t = replacement_k[:, t].view(B, H, N, 1)
        v_t           = v[:, t].view(B, H, N, 1)
        r_t           = r[:, t].view(B, H, N, 1)

        wkv_state = wkv_state * decay_t.mT - wkv_state @ removal_k_t @ (
                                        iclr_t * removal_k_t).mT
        wkv_state = wkv_state + v_t @ replacement_k_t.mT
        y = wkv_state @ r_t  # BHVK @ BHK1 = BHV1
        out[:,t] = y.view(B,C) # recombine heads

    # normalization
```

```python
    y = F.group_norm(y.view(B * T, -1), num_groups=H, params.ln_x.weight,
                                        params.ln_x.bias, eps = H * 1e-
                                        5).view(B, T, -1)

    # bonus and output
    bonus = ((r*k*params.bonus_multiplier).sum(dim=-1, keepdim=True) * v)
    bonus = bonus.view(B,T,C)    # recombine heads
    out = out + bonus
    out = (out * gate) @ params.W_output # BTC

    return out, v0, shift_state, wkv_state

def rwkv_channelmix(x, shift_state):
    x_shifted = th.cat([shift_state, x[:, :-1, :]], dim=1)
    shift_state = x[:, -1:, :]
    xk = th.lerp(x, x_shifted, params.mu_x)
    k = params.W_k @ xk
    v = params.W_v @ th.relu(k).square()
    return v, shift_state

def rwkv_model(params, input_ids, state):
    x = params.embedding(input_ids)
    x = params.layer_norm_pre(x)

    v0 = None
    layer_id = -1
    for layer in params.layers:
        layer_id = layer_id + 1
        dx, v0, state.timemix_shiftstate, state.timemix_wkvstate =
                                            rwkv_timemix(
            layer.time_mix,
            layer.layer_norm_timemix(x),
            v0,
            state.timemix_shiftstate,
            state.timemix_wkvstate,
            layer_id
        )
        x = x + dx
        dx, state.chanmix_shiftstate = rwkv_chanmix(
            layer.channel_mix,
            layer.layer_norm_chanmix(x),
            state.chanmix_shiftstate
        )
        x = x + dx

    x = params.layer_norm_out(x)
    logits = params.head(x)
    return logits, state
```

## H  PyTorch code For Naive WKV7 Kernel (Forward and Backward)

```python
class WKV7_Kernel(nn.Module):
    def __init__(self):
        super().__init__()

    def forward(self, r, w, k, v, a, b, state):
        r = r.view(B, T, H, N)
        k = k.view(B, T, H, N)
        v = v.view(B, T, H, N)
        a = a.view(B, T, H, N)
```

```
        b = b.view(B, T, H, N)
        self.state_cache = torch.zeros((B, T+1, H, N, N))
        self.state_cache[:, 0, :] = state
        w = torch.exp(-torch.exp(w.view(B, T, H, N)))
        out = torch.zeros((B, T, H, N))
        for t in range(T):
            kk = k[:, t, :]
            rr = r[:, t, :]
            vv = v[:, t, :]
            aa = a[:, t, :]
            bb = b[:, t, :]
            state = (
                state * w[: , t, :, None, :]
                + torch.einsum('bhik,bhk,bhj->bhij', state, aa, bb)
                + torch.einsum('bhj,bhi->bhij', kk, vv)
            )
            self.state_cache[:, t+1, :] = state
            out[:, t, :] = torch.einsum('bhj,bhij->bhi', rr, state)
        return out, state

    def backward(self, r, w0, k, v, a, b, gout, gstate):
        gout = gout.view(B, T, H, N)
        gr = torch.zeros((B, T, H, N))
        gw = torch.zeros((B, T, H, N))
        gk = torch.zeros((B, T, H, N))
        gv = torch.zeros((B, T, H, N))
        ga = torch.zeros((B, T, H, N))
        gb = torch.zeros((B, T, H, N))
        w = torch.exp(-torch.exp(w0.view(B, T, H, N)))
        for t in range(T-1, -1, -1):
            gr[:, t, :] = torch.matmul(
            gout[:,t,:,None,:], self.state_cache[:,t+1,:]).squeeze(-2)
            gstate.add_(torch.matmul(
            gout[:,t,:,:,None], r[:,t,:,None,:]))
            gk[:, t, :] = torch.matmul(
            v[:,t,:,None,:], gstate).squeeze(-2)
            gv[:, t, :] = torch.matmul(
            gstate, k[:,t,:,:,None]).squeeze(-1)
            ga[:, t, :] = torch.einsum(
            'bhik,bhj,bhij->bhk',
            self.state_cache[:, t, :], b[:, t, :], gstate)
            gb[:, t, :] = torch.einsum(
            'bhik,bhk,bhij->bhj',
            self.state_cache[:, t, :], a[:, t, :], gstate)
            gw[:, t, :] = torch.einsum(
            'bhij,bhij->bhj',
            self.state_cache[:, t, :], gstate)
            gstate      = torch.einsum(
            'bhj,bhij->bhij', w[:, t, :], gstate) \
            + torch.einsum(
            'bhk,bhj,bhij->bhik', a[:, t, :], b[:, t, :], gstate)
        gw = -torch.exp(w0-torch.exp(w0)) * gw
        return gr, gw, gk, gv, ga, gb, gstate
```

## I    Speed and Memory Usage Details

We compare the training speed and memory usage of the RWKV-7 attention-like kernel with the RWKV-6 kernel and Flash Attention v3 (Shah et al., 2024). The "RWKV-7" kernel accelerates bfloat16 matrix multiplications with modern CUDA instructions. We also include the "RWKV-7 fp32" kernel, which is simpler and performs all its internal calculations using float32. Although the bfloat16 kernel is faster, to maximize precision, the RWKV-7 fp32 kernel was used to train the RWKV-7 World models.

Our CUDA kernels are tuned for head dimension 64, as used in the RWKV-7-World models. The kernels still perform well for head dimension 128, but their efficiency drops off at larger head dimensions. There exist other RWKV-7 implementations which focus on head dimensions greater than 128. A key example is the Flash Linear Attention library (Yang & Zhang, 2024), which offers a Triton-based implementation designed for these larger configurations.

## I.1 Speed

In Figure 3, we time the forward + backward pass of each kernel for batch size 8, head dimension 64 and model dimension 4096 (64 wkv heads) on an H100 SXM GPU, for varying sequence lengths. Although Flash Attention v3 is heavily optimized for the H100 GPU, it scales quadratically with sequence length, while the RWKV models scale linearly. This makes the RWKV models faster than attention for large sequence lengths. Furthermore, the optimized RWKV-7 kernel is about three times faster than the official RWKV-6 kernel.

The forward pass of RWKV-7 is about twice as fast as the backward pass. For inference, the forward pass does not need to store the wkv state, making it faster. For example, for sequence length 16k, the forward pass without storing state takes 7.9 ms, while the forward pass with storing state takes 11.2 ms, the backward pass takes 22.5 ms, and the Flash Attention v3 forward pass takes 33.9 ms.

## I.2 Memory

The peak training memory usages of the tested models are well described by the formulas derived below. For example, the runs in Figure 3 required peak memory within 2% of the estimates.

In the tested kernels, the memory required per stored variable (e.g. q, k, or v in attention) is

$$\text{batch size} \times \text{model dimension} \times \text{sequence length} \times 2 \text{ bytes for bfloat16.}$$

For sequence length 1024, this is 64MB per variable. To calculate memory usage, we may use Flash Attention v3: 10 variables, RWKV-6: 10 variables, RWKV-7: 18 variables, RWKV-7 fp32: 24 variables.

Flash Attention v3 requires 4 variables q, k, v and output for the forward pass, and the corresponding 4 gradients. Finally, the backward pass uses a temporary variable to accumulate the gradient of q in float32, yielding 2 variables worth of addition memory, for a total of 4+4+2 = 10.

RWKV-6 requires 5 variables r, w, k, v, and output for the forward pass and also the corresponding 5 gradients for the backward pass, for a total of 10.

RWKV-7 uses 7 variables in the forward pass (r, w, k, v, $-\hat{\kappa}$, $\hat{\kappa} \odot a$, and output), and the corresponding 7 gradients. Additionally, it stores the wkv state every 16 timesteps. At head size 64, the state contributes the equivalent of 4 variables, for a total of 18. "RWKV-7 fp32" has the same 14 forward variables and gradients, but uses more memory to store the states in float32, for a total of 24 variable equivalents.

Memory usage is constant for single token inference and follows the formulas above, minus the gradients and state storage. Pre-fill can easily be accomplished in a chunked manner, with memory usage growing linearly with regard to chunk size. This allows an easy trade-off for fast parallelized pre-fill, with user selectable maximum memory usage.

# J Extended Evaluations

## J.1 Active Parameters versus Average Benchmark Accuracy

In Figure 7 we plot active parameters versus accuracy averaged across a series of multilingual and English benchmarks.

## J.2 Associative Recall

Associative recall (AR) (Arora et al., 2023) evaluates the ability of the model to recall previously encountered information within a given context. Research indicates that a model's capacity for AR can reflect its effectiveness in learning from context (Elhage et al., 2021; Olsson et al., 2022).

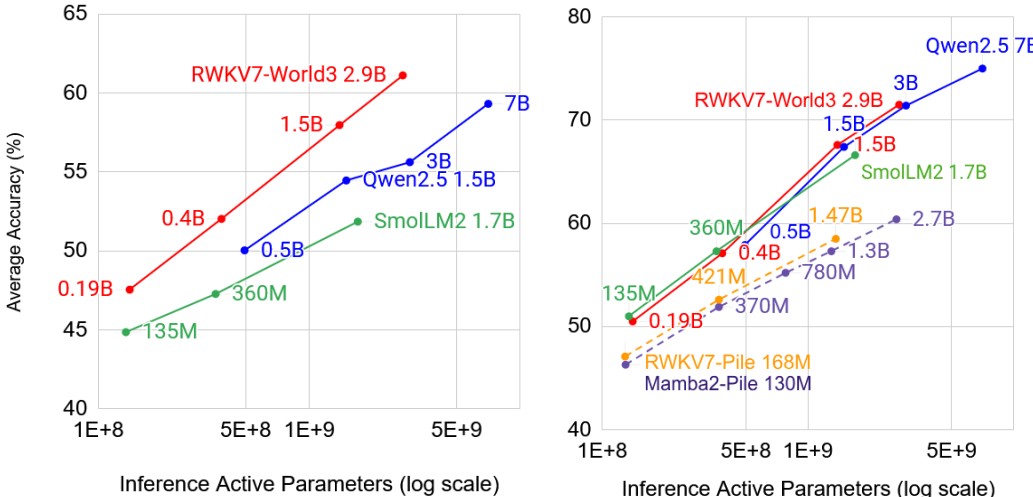

Figure 7: Active params vs. average benchmark accuracy for Multilingual (left) and English (right). Multilingual benchmarks omit Pile models due to a lack of multilingual training data in the Pile.

Consequently, AR has become a standard benchmark for developing new architectural designs in language models (Fu et al., 2023; Poli et al., 2023; Lutati et al., 2023).

In Table 14 we train two-layer RWKV-7 with MQAR, scaling the sequence length up to 2048, with RWKV-7 specific initialization, weight decay 0.1 on weight matrices, and AdamW $\epsilon$ of $1 \times 10^{-18}$ for stabilization. Interestingly, with only a WKV size of 8192, RWKV-7 is able to re-

| Dim | WKV state dim | $(64, 4)$ | $(128, 8)$ | $(256, 16)$ | $(512, 64)$ | $(1024, 128)$ | $(2048, 256)$ |
|-----|---------------|-----------|------------|-------------|-------------|----------------|----------------|
| 64  | 8192          | ✓         | ✓          | ✓           | 98.43       | 95.01          | 72.93          |
| 128 | 16384         | ✓         | ✓          | ✓           | ✓           | ✓              | 94.97          |
| 256 | 32768         | ✓         | ✓          | ✓           | ✓           | ✓              | 98.97          |
| 512 | 65536         | ✓         | ✓          | ✓           | ✓           | ✓              | ✓              |

Table 14: RWKV-7 MQAR test results. $(a, b)$ denotes sequence length and KV pair count, respectively. ✓ means over 99% accuracy. Results maxed over 3 different learning rate settings.

call 72.93% at the setting of 256 Key-value pairs. This suggests that a total of roughly $256 \times 0.7293 \times \log_2(\text{number of key tokens} \times \text{number of value tokens}) = 186.2 \times 2 \times \log_2(4096) = 4480.8$ bits of information, is stored in a 8192 dimensional state, yielding an information density of 0.547 bits per dimension.

### J.3 Fresh Internet Text Evaluation (extended)

We collected new data created after January 2025, including: newly submitted computer science and physics papers on arXiv, newly created Python/C++ open-source repositories on GitHub, recently published Wikipedia entries, new fiction on Archive of Our Own (Various, 2025), and recent news articles. Inspired by Delétang et al. (2024); Li et al. (2024b), we used compression rate as our evaluation metric. See Table 15 for details.

Remarkably, despite being trained on significantly less data than other top models, RWKV-7 Goose showed competitive performance on this temporally novel data.

### J.4 Long Context Experiments

To evaluate the ability of RWKV models to retain information over long sequences, we measured loss versus sequence position (we select tokens in range $[L/2 - 16384, L/2 + 16384]$ for document length $L$) on the PG19 test set (Rae et al., 2019) for two types of RWKV7 models and their predeces-

| Model | arXiv CS ↓ | arXiv Phys. ↓ | Github Python ↓ | Github C++ ↓ | AO3 Eng ↓ | BBC news ↓ | Wiki Eng ↓ | average ↓ |
|---|---|---|---|---|---|---|---|---|
| **Qwen2.5-1.5B** | **8.12** | 8.65 | **4.42** | **4.40** | 11.76 | 9.58 | 9.49 | **8.06** |
| RWKV-7 1.5B | 8.25 | 8.77 | 5.57 | 5.29 | **10.93** | **9.34** | **8.97** | 8.16 |
| Llama-3.2-1B | 8.37 | 8.76 | 5.18 | 5.16 | 11.69 | **9.34** | 9.07 | 8.23 |
| SmolLM2-1.7B | 8.38 | 9.04 | 5.17 | 4.94 | 11.20 | 9.40 | 9.46 | 8.23 |
| Index-1.9B | 8.34 | **8.59** | 5.65 | 5.29 | 11.49 | 9.51 | 9.23 | 8.30 |
| stablelm-2-1.6b | 8.58 | 9.08 | 5.54 | 5.45 | 11.42 | 9.24 | 9.06 | 8.34 |
| RWKV-6 1.5B | 8.62 | 9.00 | 6.06 | 5.80 | 11.09 | 9.57 | 9.30 | 8.49 |
| RWKV-5 1.5B | 8.77 | 9.11 | 6.20 | 5.92 | 11.25 | 9.75 | 9.50 | 8.64 |
| mamba2-1.3b | 8.74 | 8.74 | 6.32 | 5.71 | 11.63 | 9.74 | 9.86 | 8.68 |
| MobileLLM-1.5B | 8.82 | 9.29 | 6.79 | 6.29 | 11.59 | **9.15** | 9.22 | 8.73 |
| mamba-1.4b-hf | 8.88 | 8.86 | 6.43 | 5.81 | 11.70 | 9.83 | 9.97 | 8.78 |
| Zamba2-1.2B | 8.57 | 9.21 | 6.91 | 7.08 | 11.39 | 9.38 | 9.26 | 8.83 |
| SmolLM-1.7B | 8.38 | 9.02 | 5.76 | 6.55 | 12.68 | 9.85 | 9.89 | 8.88 |
| MobileLLM-1B | 9.03 | 9.57 | 7.03 | 6.53 | 11.86 | 9.35 | 9.43 | 8.97 |
| RWKV-4 1.5B | 9.34 | 9.80 | 6.54 | 6.16 | 11.33 | 10.00 | 9.82 | 9.00 |
| pythia-1.4b-v0 | 9.12 | 9.20 | 6.79 | 6.15 | 12.19 | 10.20 | 10.43 | 9.15 |
| Falcon3-1B-Base | 8.60 | 9.20 | 6.92 | 7.16 | 13.04 | 10.45 | 10.75 | 9.45 |
| **Llama-3.2-3B** | **7.78** | **8.10** | 4.15 | 4.59 | 10.90 | **8.70** | **8.28** | **7.57** |
| Qwen2.5-3B | 7.79 | 8.25 | **4.15** | **4.12** | 11.23 | 9.15 | 8.96 | 7.66 |
| RWKV-7 2.9B | 7.90 | 8.34 | 5.16 | 4.88 | **10.48** | 8.92 | 8.47 | 7.74 |
| stablelm-3b-4e1t | 8.15 | 8.50 | 5.28 | 4.85 | 10.89 | 8.82 | 8.51 | 7.86 |
| Minitron-4B-Base | 8.09 | 8.70 | 5.13 | 4.74 | 11.05 | 9.08 | 8.90 | 7.96 |
| recurrentgemma-2b | 8.24 | 8.52 | 5.22 | 4.80 | 11.30 | 8.94 | 8.88 | 7.99 |
| RWKV-6 3B | 8.27 | 8.58 | 5.66 | 5.39 | 10.67 | 9.17 | 8.82 | 8.08 |
| gemma-2-2b | 8.39 | 8.81 | 5.36 | 5.01 | 11.35 | 8.90 | 9.03 | 8.12 |
| mamba2attn-2.7b | 8.33 | 8.29 | 5.78 | 5.22 | 11.13 | 9.28 | 9.26 | 8.18 |
| RWKV-5 3B | 8.42 | 8.70 | 5.78 | 5.51 | 10.83 | 9.36 | 9.00 | 8.23 |
| mamba2-2.7b | 8.43 | 8.37 | 5.93 | 5.34 | 11.21 | 9.37 | 9.38 | 8.29 |
| Zamba2-2.7B | 8.17 | 8.70 | 6.30 | 6.39 | 10.97 | 8.95 | 8.74 | 8.32 |
| mamba-2.8b-hf | 8.57 | 8.52 | 6.03 | 5.46 | 11.31 | 9.49 | 9.53 | 8.41 |
| RWKV-4 3B | 8.90 | 9.27 | 6.07 | 5.67 | 10.90 | 9.57 | 9.30 | 8.53 |
| pythia-2.8b-v0 | 8.72 | 8.73 | 6.29 | 5.71 | 11.66 | 9.74 | 9.82 | 8.67 |

Table 15: Compression rate (unit: %) compared across different language models on various data sources, including arXiv papers, GitHub repositories, AO3 fiction, and news articles created after January 2025. We define compression rate as: $(log_2(e) * \text{average loss over document} * \text{document tokens})/(8 * \text{document length in bytes})$.

sors trained on either The Pile dataset or World dataset. Despite sharing the same architecture and being pretrained on 4k context windows, models trained on different datasets exhibited different behaviors. The Pile-trained RWKV7 showed more significant loss reduction on long contexts compared to its predecessors, demonstrating effective long-context extrapolation (see Figure 8). Surprisingly, for RWKV7 trained on the World dataset, when processing contexts longer than 10k, the loss began to show an increasing trend (see Figure 9). We speculate this is because the larger dataset and model size created inductive biases that caused overfitting to specific context lengths. Further experiments showed that fine-tuning on long contexts can restore its long context capabilities.

To further test RWKV-7 long-context retrieval abilities, we conduct a pass-key retrieval evaluation following the approach of (Chen et al., 2024b) and plot the results in Figure 10. In this evaluation, a single sentence is repeated multiple times within a long context window, with a key phrase embedded at different positions. RWKV7-World3-1.5B achieves perfect accuracy up to a context length of 19600 tokens but exhibits degradation beyond 20600 tokens. The larger RWKV7-World3-2.9B extends perfect retrieval up to 32000 tokens, highlighting the benefits of scaling. However, performance begins to degrade beyond this point.

To explore potential improvements, we fine-tuned RWKV7-World3-1.5B and RWKV7-World3-2.9B on packed training sequences of length 128k tokens from a specially constructed dataset, which leads to further improvements in retrieval accuracy. With this fine-tuning, RWKV-7 (1.5B) reliably retrieves key phrases up to 29k tokens, and degradation is observed only around 40k tokens. RWKV-7 (2.9B) reliably retrieves the pass key up to 30k tokens, and degrades around 50k tokens.

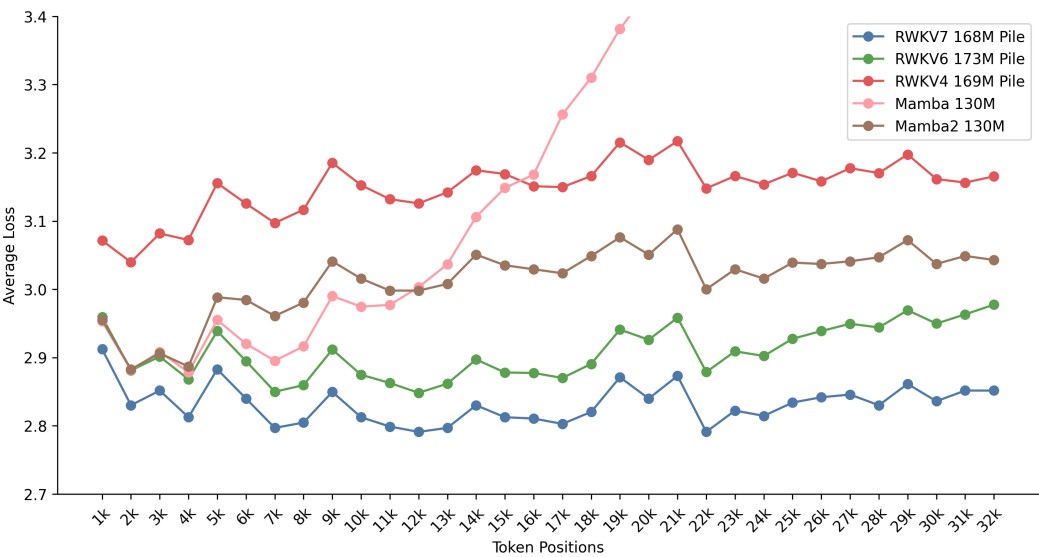

Figure 8: PG19 loss versus sequence position for RWKV and Mamba models trained on The Pile datasets.

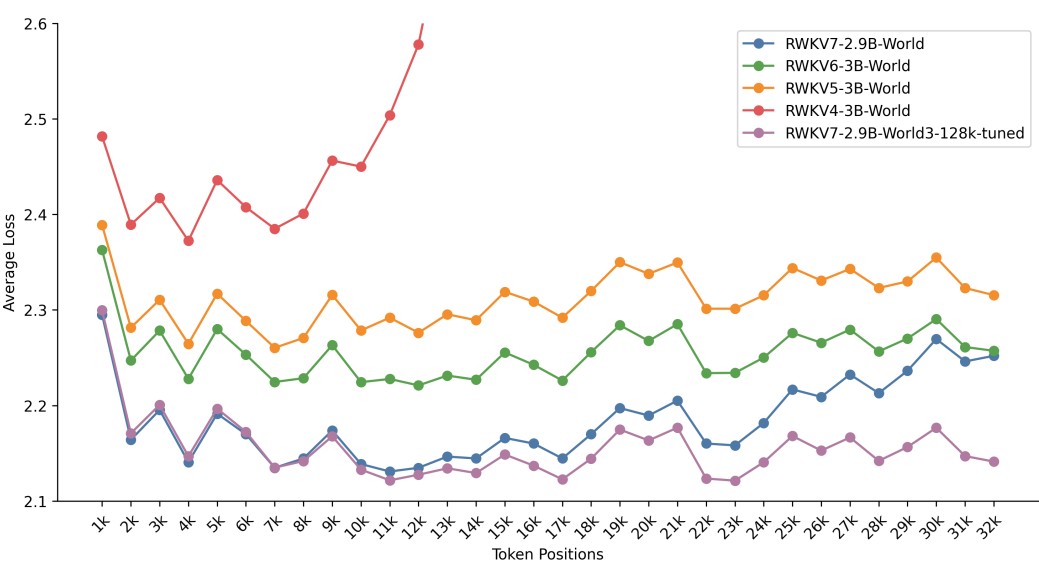

Figure 9: PG19 loss versus sequence position for RWKV7 models and predecessors trained on the World dataset.

Our context length extension dataset is comprised of both public and custom sources listed in Table 16. We employed a document length-based weighting scheme to prioritize longer contexts during training. To approximate document lengths of 128,000 tokens, we used character counts of 512,000. Documents of less than 32,768 characters were assigned a weight of 1.0, while longer documents were assigned linearly increasing weights between 2.0 and 3.0, with a cap of 3.0 beyond 512,000. This method increases the inclusion of longer documents to bolster the model's handling of extended contexts while retaining shorter documents for diversity.

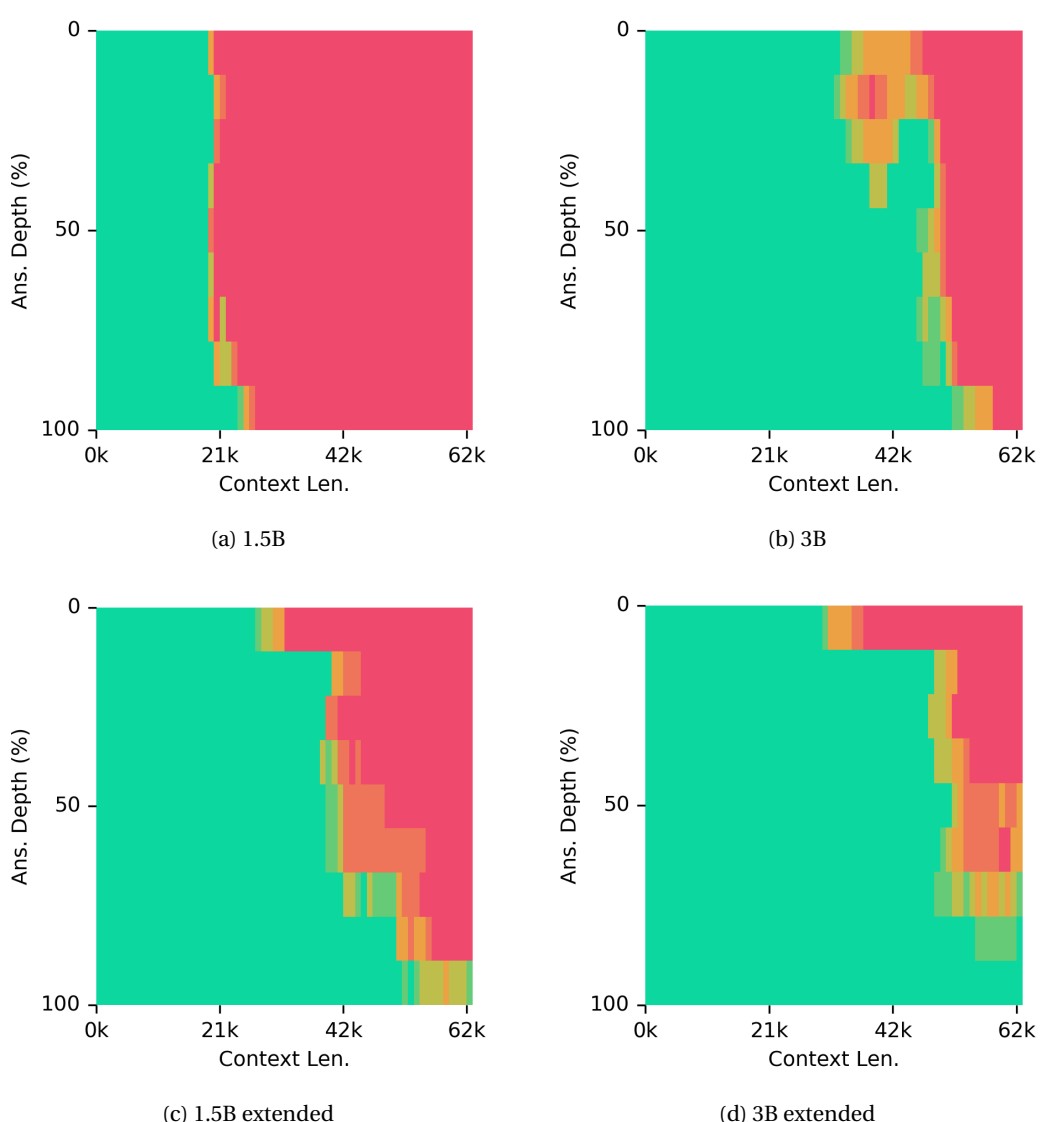

Figure 10: RWKV7-World3 pass-key retrieval evaluation

| Dataset | Type | Amount |
|---|---|---|
| dclm-baseline-1.0 | Public | 25% |
| fineweb-edu | Public | 15% |
| fineweb | Public | 5% |
| codeparrot/github-code | Public | 10% |
| arXiv-CC0-v0.5 | Custom | 10% |
| SuperWikiNEXT-32B | Custom | 10% |
| public domain books | Custom | 15% |
| the-stack (filtered) | Custom | 10% |

Table 16: Context length extension dataset components

## J.5 Evaluating State Tracking Using Group Multiplication

We adopt the experimental setting from Merrill et al. (2024) to evaluate the state-tracking capabilities of RWKV7 in comparison to Transformer, Mamba, S4, and classical RNN models. Given a

sequence $g_0, g_1, g_2, \ldots, g_n$ drawn from $A_5$, $A_4 \times \mathbb{Z}_5$, or $\mathbb{Z}_{60}$, each step $i$ is labeled with the cumulative product of the first $i$ elements.

We plot the minimum number of layers required to achieve over 95% validation accuracy on group multiplication tasks, as a function of sequence length and group structure. The results are shown in Figure 11. Our findings indicate that RWKV-7 exhibits stronger state-tracking capabilities than Transformers, Mamba, and S4, though slightly weaker than classical RNNs. Figure 11 also aligns with our theory from Appendix D.2, which predicts that RWKV-7 can perform state tracking and recognize any regular language with a constant number of layers. RWKV-7 has no expressivity advantage for state tracking compared to classical RNNs, which can recognize any regular language in a single layer. However, classical RNNs, while being theoretically expressive, typically suffer from gradient vanishing and memorization problems (Zucchet & Orvieto, 2024) and cannot be parallelized efficiently, unlike RWKV-7.

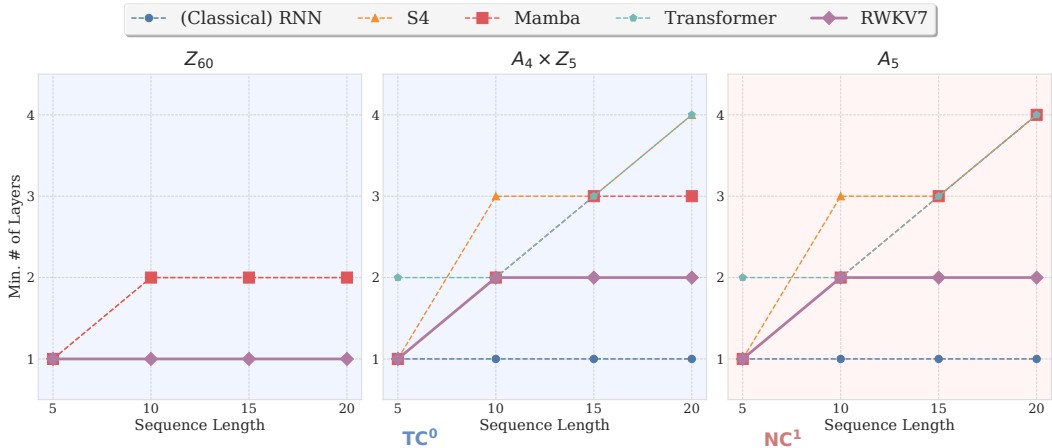

Figure 11: Minimum number of layers (lower is better) required to attain > 95% validation accuracy on group multiplication problems by sequence length and group.

## K   Multimodal Experiments

In this section, we explore the capabilities of Goose when extended to handle multimodal tasks, where the model processes and integrates textual inputs with inputs from a different domain.

### K.1   RWKV for Image Understanding

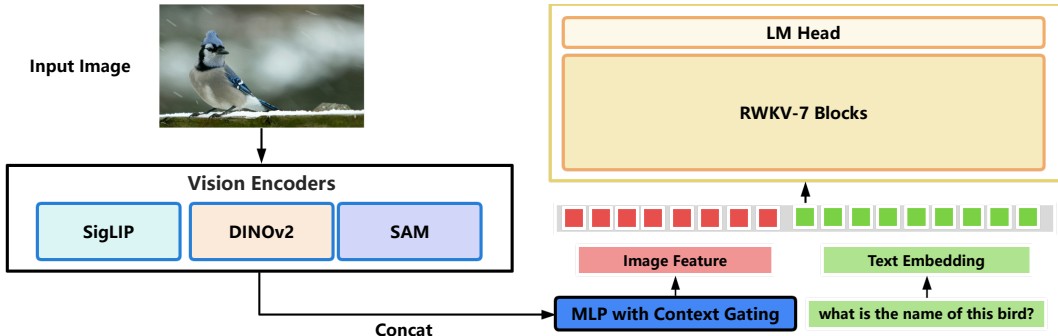

Figure 12: The architecture of VisualRWKV-7. The input image is processed by three vision encoders, and the obtained features are concatenated. Afterward, they are projected through an MLP with context gating to align with the dimensions of the RWKV-7 block. Finally, the image features are concatenated with the text embeddings and fed into the RWKV-7 LLM.

To demonstrate the modeling capabilities of RWKV-7, we constructed VisualRWKV-7 (Figure12), a visual language model based on the RWKV-7 block, to evaluate the image understanding capabilities of RWKV-7. VisualRWKV-6 (Hou et al., 2024) used the CLIP encoder, which focused on processing low-resolution images and achieved good results. VisualRWKV-7 replaces the CLIP encoder with SigLIP and DINO visual encoders, and introduced a new high-resolution SAM vision encoder, which enhances the model's supported resolution to 1024 x 1024.

| Method | Vision Encoder | LLM | VQA | SQA | TQA | GQA |
|--------|---------------|-----|-----|-----|-----|-----|
| VisualRWKV-6 | SigLIP+DINOv2+SAM | RWKV6-1.6B | 73.6 | 57.0 | 48.7 | 58.2 |
| VisualRWKV-6 | SigLIP+DINOv2+SAM | RWKV6-3.1B | 79.1 | 62.9 | 52.7 | 61.0 |
| VisualRWKV-7 | SigLIP+DINOv2+SAM | RWKV7-0.1B | 75.2 | 50.6 | 37.9 | 59.9 |
| VisualRWKV-7 | SigLIP+DINOv2+SAM | RWKV7-0.4B | 77.9 | 55.0 | 41.1 | 62.3 |
| VisualRWKV-7 | SigLIP+DINOv2+SAM | RWKV7-1.5B | 79.8 | 59.7 | 49.5 | 63.2 |
| VisualRWKV-7 | SigLIP+DINOv2+SAM | RWKV7-2.9B | **80.5** | **63.4** | **58.0** | **63.7** |

Table 17: A comparison of VisualRWKV-7 to other Visual Language Models across 4 distinct benchmarks. We evaluate these models on benchmarks: GQA(Hudson & Manning, 2019), SQA(Lu et al., 2022), TQA(Singh et al., 2019) and VQA(Li et al., 2023b).

The experimental results of VisualRWKV-7 are shown in Table 17. The vision encoders used in both VisualRWKV-7 and VisualRWKV-6 are identical, and the training data remains consistent, aligned with the training data of LLaVA-1.5. The first stage consists of 558k alignment data, while the second stage includes 665k SFT data.

VisualRWKV-7 0.1B and 0.4B outperform VisualRWKV-6 1.6B on the in-domain benchmarks VQAv2 and GQA and rapidly approach VisualRWKV-6 1.6B on two other benchmarks. The experimental results are highly compelling. With only 1/4 of the parameters (1.6B vs. 0.4B), VisualRWKV-7 surpasses VisualRWKV-6 on the VQAv2 and GQA benchmarks, demonstrating the powerful modeling capabilities of RWKV-7.

On the out-of-domain benchmark SQA, VisualRWKV-7 2.9B also outperforms VisualRWKV-6 3.1B, indicating that VisualRWKV-7 possesses strong generalization ability. In the TextQA (TQA) benchmark, which assesses a model's associative recall, VisualRWKV-7 2.9B achieves a 5.3-point improvement over VisualRWKV-6 3.1B, further proving its superior associative recall capabilities.

### K.2 RWKV for Audio Modeling

To investigate the effectiveness of RWKV-7 for audio modeling, we introduce AudioRWKV-7, a novel adaptation of RWKV-7 for audio embedding analysis. We use a bi-directional modification to RWKV-7, similar to the modification used by Duan et al. (2024). We employ this approach to interpret and process complex, high-dimensional spectrogram features. To further capture acoustic and temporal characteristics, an mel-spectrogram is divided into patch tokens using a Patch-Embed CNN with a kernel size of $(P \times P)$ and sequentially fed into the model. The width and height of an audio mel-spectrogram represent the time and frequency bins, respectively. Typically, the time dimension is significantly longer than the frequency dimension. To effectively capture relationships among frequency bins within the same time frame, the mel-spectrogram is first segmented into patch windows $w_1, w_2, ..., w_n$, followed by further division of patches within each window. The token sequence follows the order: *time → frequency → window*. This arrangement ensures that patches corresponding to different frequency bins within the same time frame are positioned adjacent to each other in the input sequence.

We evaluated the performance of AudioRWKV-7 across multiple model scales and architectures, using the AudioSet dataset (Gemmeke et al., 2017). Detailed experimental outcomes are presented in Table 18. From the results we find that our model achieves comparable performance with a much smaller parameter count when compared with CNN, Transformer and Mamba based architectures, and exceeds the performance of AudioRWKV-6. These findings demonstrate the robustness and versatility of AudioRWKV. Note that to ensure a fair comparison, we retrained AudioRWKV-6 without ensembling models with different patch settings, which accounts for the difference in results from (Peng et al., 2024a).

| Model | #Parameters | Architecture | mAP ↑ |
|---|---|---|---|
| DeepRes (Ford et al., 2019) | 26M | CNN | 0.392 |
| HST-AT | 88.5M | Transformer | 0.433* |
| HST-AT pretrained(Chen et al., 2022) | 88.5M | Transformer | 0.429* |
| MambaOut(Yu & Wang, 2024) | 101.3M | Mamba | 0.397 |
| AudioRWKV-6 | 8.9M | RWKV6 | 0.381 |
| AudioRWKV-6 | 19.8M | RWKV6 | 0.426 |
| AudioRWKV-7 | 8.9M | RWKV7 | 0.392 |
| AudioRWKV-7 | 19.8M | RWKV7 | 0.431 |

Table 18: A comparison of mean Average Precision (mAP) among AudioRWKV7 and other baselines on AudioSet dataset. HST-AT pretrained is a variation that uses vision transformer to initialize the weights. *Results reproduced by ourselves.

## L  Board Game Modeling

Previous research (Schultz et al. (2024); Topsakal et al. (2024)) has shown that board games can serve as potentially effective tools for evaluating a model's capabilities. As an RNN with powerful state tracking abilities, RWKV-7 is highly suitable for board game modeling and conducting extended searches directly within context to find better strategies. As an early exploration, we tested RWKV-7's board game modeling capabilities and its ability to perform in-context search on the game of Othello (Reversi). We found an expanded formulation of RWKV-7 to be useful for this task, which we designate as RWKV-7a: $S_t = S_{t-1} \text{diag}(w_t)(I - c\hat{\kappa}_t^T(a_t \odot \hat{\kappa}_t)) + v_t^T k_t$ This formula allows the full range of (-1,1) eigenvalues when $c = 2$.

**Data**  We designed training data that guides the model to predict legal moves, evaluate these moves, and perform Alpha-Beta pruning. Each training sample consists of three components:

- **Input section**: Captures the game state, including the current board position, active player, and search parameters (tree depth and width settings).
- **Reasoning section**: Varies with search settings:
  - Without search (depth or width = 1): Lists legal moves and their evaluations
  - With search (depth and width > 1): Performs Alpha-Beta pruning to find optimal moves

  All move evaluations in the reasoning section were generated using the Egaroucid engine (Yamana, 2025).
- **Output section**: This contains the final move decision and displays the resulting board position after implementing the move.

An example of our training data is shown in sample 1.

Code Listing 1: Training Data Example

```
<input>
. . o o * * . .
* . o o o * * .
* * * o * * o o
* * * * * o o o
* o * o o * o o
* * o o o * o .
* . o o o o . .
. . . . o o . .
NEXT o
MAX_WIDTH-2
MAX_DEPTH-2
</input>
<reasoning>
Possible moves and score: g1 -19 h1 -01 b2 -08 h2 -23 b7 -12 g7 -09
<stack>
```

Remaining_Depth:2
Max_Node Alpha: −in Beta: +in Best: −− Current: h1 −01 Unexplored: b2 −08
</stack>
=> Search next node
[Depth limit not reached]
<board>
. . o o * * . o
* . o o o * o .
* * * o * o o o
* * * * o o o o
* o * o o * o o
* * o o o * o .
* . o o o o . .
. . . . o o . .
</board>
NEXT *
Possible moves and score: b1 +02 b2 +05 h2 +10 h6 +03 h7 +08 c8 +06 d8 +01 g8 +09
[Current player has legal moves]
[Internal node – expand]
<stack>
Remaining_Depth:1
Min_Node Alpha: −in Beta: +in Best: −− Current: d8 +01 Unexplored: b1 +02
Max_Node Alpha: −in Beta: +in Best: −− Current: h1 −01 Unexplored: b2 −08
</stack>
=> Search next node
[Depth limit reached – evaluate all leaves]
[Updated stack]
<stack>
Remaining_Depth:1
Min_Node Alpha: −in Beta: +01 Best: d8 Current: −− −−− Unexplored:
Max_Node Alpha: +01 Beta: +in Best: h1 Current: b2 −08 Unexplored:
</stack>
=> Search next node
[Depth limit not reached]
<board>
. . o o * * . .
* o o o o * * .
* o o o * * o o
* o * o * o o o
* o * o o * o o
* * o o o * o .
* . o o o o . .
. . . . o o . .
</board>
NEXT *
Possible moves and score: a1 −07 b1 +13 h6 −01 b7 −08 g7 +08 h7 −02 c8 +01 d8 −03 g8 +04
[Current player has legal moves]
[Internal node – expand]
<stack>
Remaining_Depth:1
Min_Node Alpha: +01 Beta: +in Best: −− Current: b7 −08 Unexplored: a1 −07
Max_Node Alpha: +01 Beta: +in Best: h1 Current: b2 −08 Unexplored:
</stack>
=> Search next node
[Depth limit reached – evaluate all leaves]
[Updated stack]
<stack>
Remaining_Depth:1
Min_Node Alpha: +01 Beta: −08 Best: b7 Current: −− −−− Unexplored:

```
Max_Node Alpha: +01 Beta: +in Best: h1 Current: -- --- Unexplored:
</stack>
[End of search]
> Playing h1
</reasoning>
<output>
 h1
. . o o * * . o
* . o o o * o .
* * * o * o o o
* * * * o o o o
* o * o o * o o
* * o o o * o .
* . o o o o . .
. . . . o o . .
</output>
```

**Training**   We trained RWKV-7 models with 9M and 26M parameters respectively on 6 million samples. By tracking the loss across different token types during training (figure 13), we noticed that the model first mastered output formatting, then developed board state tracking capability, and continuously improved its evaluation accuracy throughout training.

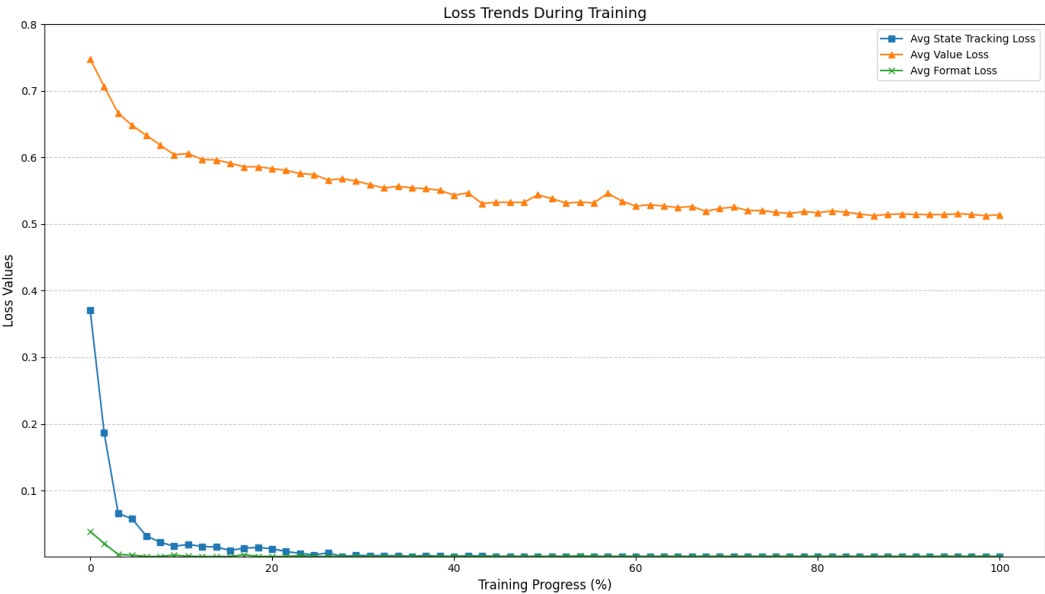

Figure 13: Reversi training loss for different token types

**Evaluation**   We control the model's thinking budget by setting the width and depth of Alpha-beta pruning, and test the win rate against baseline a model (depth=1, width=1) under different budgets. As shown in figure 14, by increasing the testing budget, RWKV-7 can effectively search for better strategies, demonstrating a positive test-time scaling law on this task.

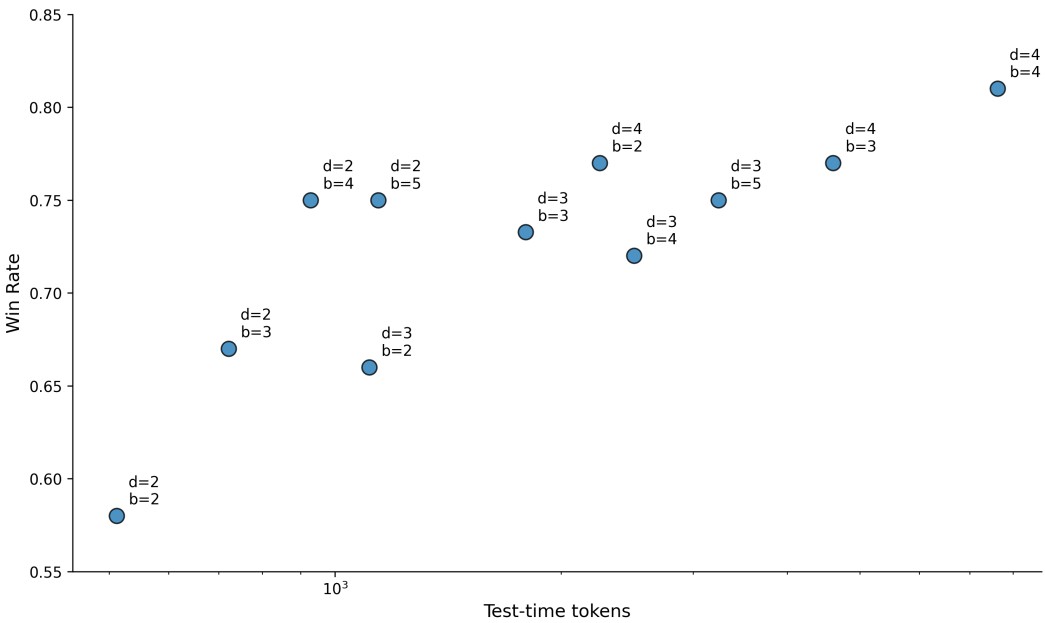

Figure 14: Reversi token consumption and win rates under different search configurations

## M  Ablation Experiments

### M.1  The Pile

To demonstrate the architectural advantages of RWKV-7, we conducted ablation experiments by training models of three different sizes—168M, 421M, and 1.47B parameters—on the full Pile dataset (Gao et al., 2020).

| Model | Tokens (B) | lmb.o ppl ↓ | lmb.o acc ↑ | hella acc_n ↑ | piqa acc ↑ | arcE acc ↑ | arcC acc ↑ | glue acc ↑ | WG acc ↑ | sciq acc ↑ | avg acc ↑ |
|---|---|---|---|---|---|---|---|---|---|---|---|
| RWKV4-169M-Pile | 332 | 29.2 | 33.2 | 32.2 | 64.8 | 47.1 | 19.9 | 47.6 | 51.2 | 77.6 | 46.7 |
| Pythia-160M | 300 | 37.3 | 35.4 | 30.3 | 62.3 | 43.6 | 19.5 | 46.5 | 51.3 | 75.4 | 45.5 |
| Mamba-130M | 300 | 16.0 | 44.3 | 35.3 | 64.5 | 48.0 | 19.7 | 48.5 | 52.1 | 78.2 | 48.8 |
| Mamba2-130M | 300 | 16.8 | 43.9 | 35.3 | 64.9 | 47.4 | **20.9** | 45.8 | **52.6** | 81.0 | 49.0 |
| RWKV6-173M-Pile | 332 | 16.0 | 44.5 | 34.9 | 64.4 | **48.3** | 19.7 | 48.9 | 51.9 | 80.6 | 49.2 |
| RWKV7-168M-Pile | 332 | **14.2** | **45.7** | **36.9** | **65.5** | 47.9 | 19.7 | **49.1** | 52.4 | **81.6** | **49.8** |
| RWKV4-430M-Pile | 332 | 13.1 | 45.1 | 40.8 | 67.7 | 52.8 | 24.1 | 49.4 | 52.0 | 80.7 | 51.6 |
| Pythia-410M | 300 | 10.8 | 51.6 | 40.6 | 66.7 | 51.9 | 21.4 | 44.1 | 53.3 | 81.5 | 51.4 |
| Mamba-370M | 300 | 8.1 | 55.6 | 46.5 | 69.5 | 55.0 | 25.0 | 46.8 | 55.5 | 84.5 | 54.8 |
| Mamba2-370M | 300 | 8.0 | 55.9 | 46.9 | **70.5** | 54.8 | **25.1** | 48.1 | 55.4 | 85.3 | 55.2 |
| RWKV7-421M-Pile | 332 | **7.2** | **57.9** | **48.0** | 69.3 | **56.3** | 23.5 | 50.3 | **56.4** | **85.9** | **56.0** |
| RWKV4-1.5B-Pile | 332 | 7.1 | 56.4 | 52.8 | 72.2 | 60.7 | 24.9 | **50.5** | 54.3 | 85.8 | 57.2 |
| Pythia-1.4B | 300 | 6.1 | 61.7 | 52.0 | 70.8 | 60.5 | 26.1 | 47.7 | 57.5 | 86.6 | 57.9 |
| Mamba-1.4B | 300 | 5.0 | 65.0 | 59.1 | **74.2** | **65.5** | 29.8 | 46.2 | 61.4 | 87.3 | 61.1 |
| Mamba2-1.3B | 300 | 5.0 | 65.6 | 59.9 | 73.2 | 64.2 | 29.9 | 46.1 | 61.0 | 89.8 | 61.2 |
| RWKV7-1.47B-Pile | 332 | **4.8** | **67.0** | **61.8** | 73.6 | 64.9 | **30.2** | 48.0 | **64.4** | **91.1** | **62.6** |

Table 19: English focused benchmarks, including LAMBADA (**lmb.o**) (Paperno et al., 2016), Hellaswag (**hella**) (Zellers et al., 2019), PIQA (Bisk et al., 2020), AI2 ARC (**arcE**, **arcC**) (Bhakthavatsalam et al., 2021), GLUE (Wang et al., 2018), Winogrande (**WG**) (Sakaguchi et al., 2021), SciQ (Welbl et al., 2017).

These results highlight the consistent improvements brought by the RWKV-7 architecture over earlier RWKV models, even when trained on the same dataset. As all RWKV models shown were trained under identical configurations and dataset, this underscores the inherent architectural

advantages of RWKV-7 over its predecessors. Notably, the performance gap sustains as the model size increases, suggesting that RWKV-7 may scale more effectively than its predecessors.

### M.2  Architecture Choice Ablations

We ran a series of ablation experiments to validate the various improvements made in RWKV-7 versus some of the more restrictive choices seen in other DeltaNet and post-DeltaNet related work. These improvements were:

- using a vector-valued decay $w$ instead of a scalar-valued decay;
- using a vector-valued in-context learning rate $a$ instead of a scalar-valued rate;
- using different removal $\kappa$ and replacement $\tilde{k}$ keys, instead of the same for both; and
- adding the bonus term $u_{t,j}v_{t,j}$ to the output step (Equation 20).

We trained a small 6-layer, $d_{model} = 768$ Goose model on the 1.6B token minipile (Kaddour, 2023) dataset at context length 512 and obtained the loss results shown in Table 20.

| Model | Training Loss | Validation Loss |
|---|---|---|
| Goose | 2.834 | 2.541 |
| Goose, scalar decay | 2.873 | 2.609 |
| Goose, scalar in-context learning rate | 2.843 | 2.591 |
| Goose, same removal/replacement keys | 2.840 | 2.560 |
| Goose, no bonus term | 2.841 | 2.588 |

Table 20: Ablation results for 6 layer 768 dimension Goose model

## N  State Inspections

We inspected the WKV matrix states of RWKV-7 and compared them with those of RWKV-5 and RWKV-6. We analyzed the following aspects:

1. Visualization example of WKV state matrices, to better understand the structure and behavior of the matrices.
2. Root Mean Square (RMS) of matrix elements, to assess the numerical stability of those WKV matrices.
3. Stable Rank (SR) of the matrices (Rudelson & Vershynin, 2007), defined as the square of (the Frobenius norm divided by the spectral norm):

$$\mathrm{SR}(A) := \left(\frac{\|A\|_F}{\|A\|_2}\right)^2.$$

This serves as a rough measure to the effective amount of information of the states.

For this analysis, we selected 10 samples from the validation set of the PG19 dataset (Rae et al., 2019), ensuring that each sample had a sequence length of at least 8192 tokens. We tested the 1.5B parameter versions of RWKV-5, RWKV-6, and RWKV-7, plotting on the appearance, stable rank and RMS values of their WKV matrices.

We observed that the WKV states of RWKV-7 had significantly smaller RMS values compared to RWKV-5 and RWKV-6. The entries of the WKV matrix in RWKV-7 were consistently of order $O(1)$ (i.e., no outliers, and does not grow over context length), whereas RWKV-5 and RWKV-6 constantly produced outliers on the order of thousands (see Figures 15 and 16). This indicates that RWKV-7 has better numerical stability during training and inference.

Interestingly, the stable rank of the WKV matrix in RWKV-7 has shown to be lower than that of RWKV-5 and RWKV-6 for context longer than 32. A lower stable rank typically suggests that the matrix contains less information or has a more compressed representation. However, this observation appears to contradict the experimental results showing that RWKV-7 performs better on tasks requiring long-term memory. We hypothesize that this contradiction can be explained by RWKV-7's enhanced state evolution mechanism, which enables RWKV-7 to achieve stronger information

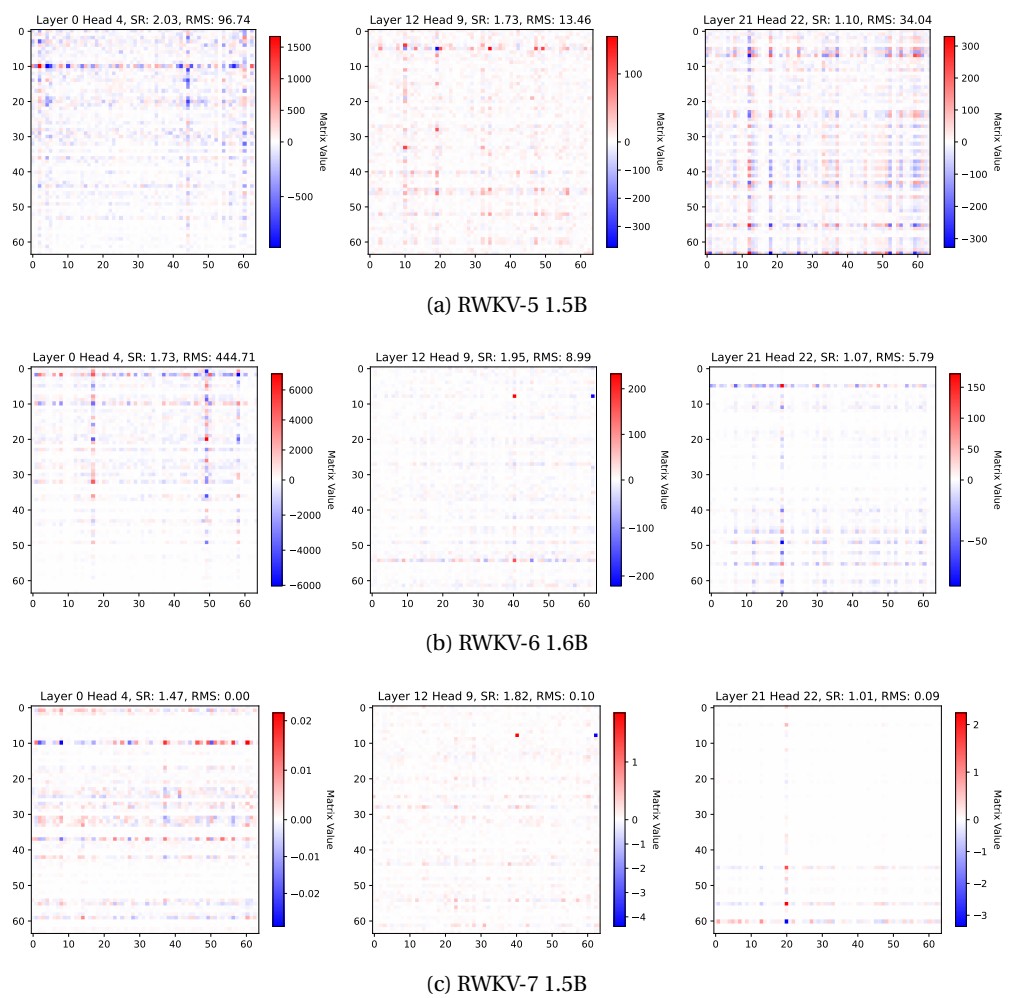

Figure 15: Visualization example of RWKV's WKV matrices.

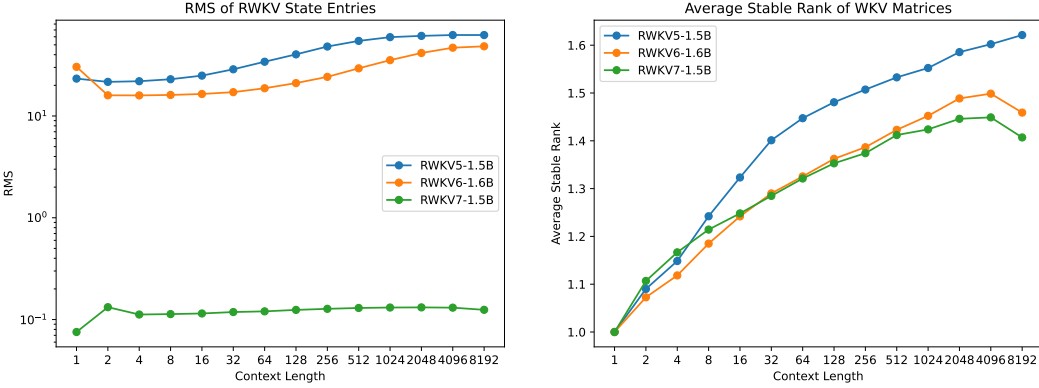

Figure 16: The global RMS and average stable rank of WKV matrices, plotted over sequence length.

compression and utilization capabilities and allowing it to maintain important information in a more compact form. This dual capability likely contributes to both the reduced stable rank and the improved performance on memory-intensive tasks.

## O  Parameters Statistics

In Section 4, the actual ranges and statistical metrics of the parameters within the trained model are not specified. To facilitate a better understanding of the role of these parameters in the practical models, this appendix provides empirical statistical metrics of selected parameters from the released RWKV-7 model.

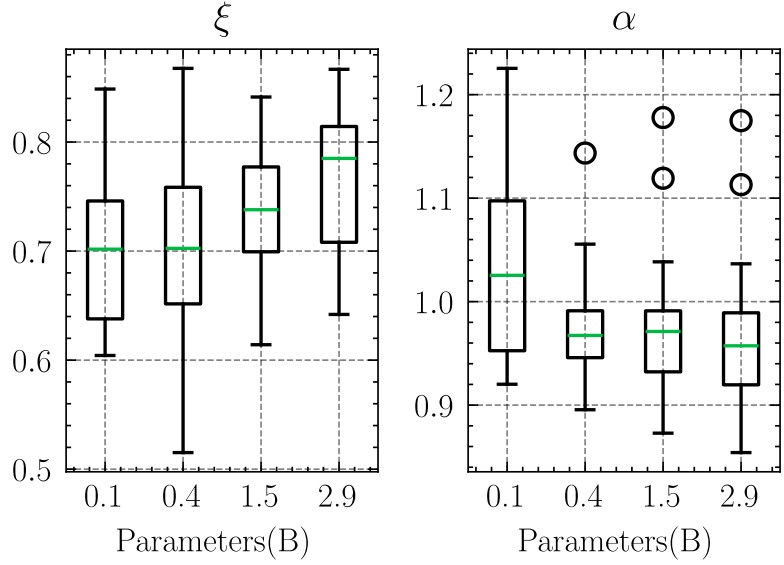

Figure 17: Box plots of $\xi$ and $\alpha$ across models

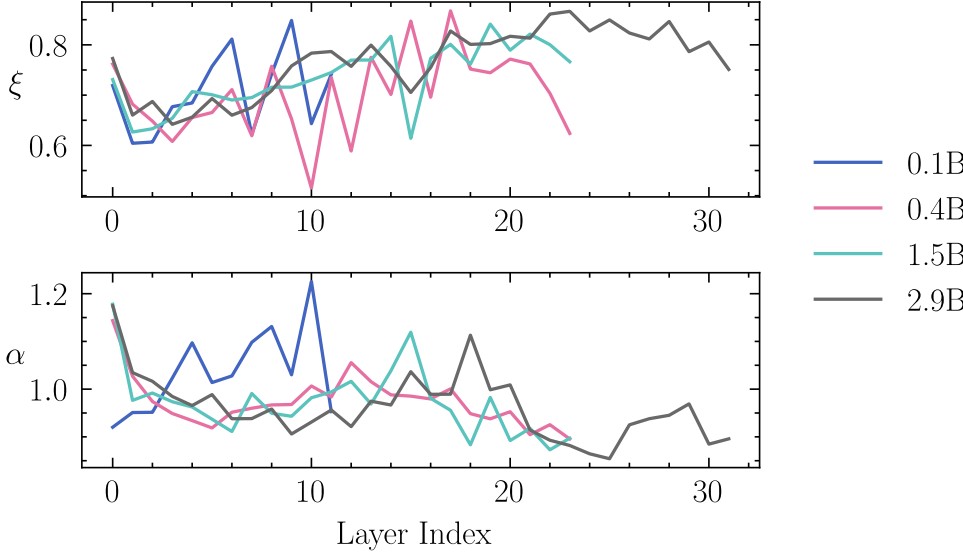

Figure 18: Mean $\xi$ and $\alpha$ across layers for different models

## P  Initial Token Sensitivity

In our evaluation of LAMBADA (Paperno et al., 2016), we found that RWKV-7's performance varies significantly under different settings. After ruling out precision issues, we investigated the

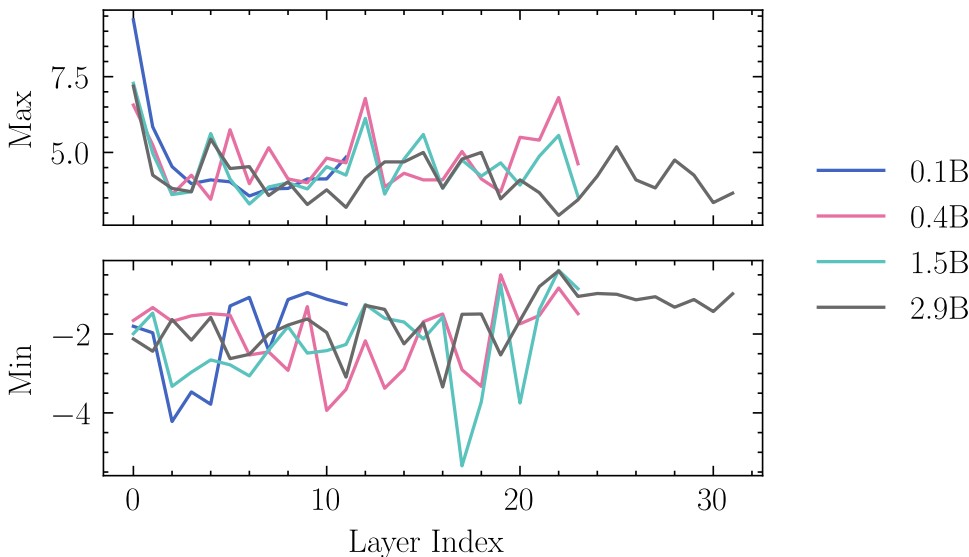

Figure 19: Maximum and minimum of $\xi$ across layers in different models

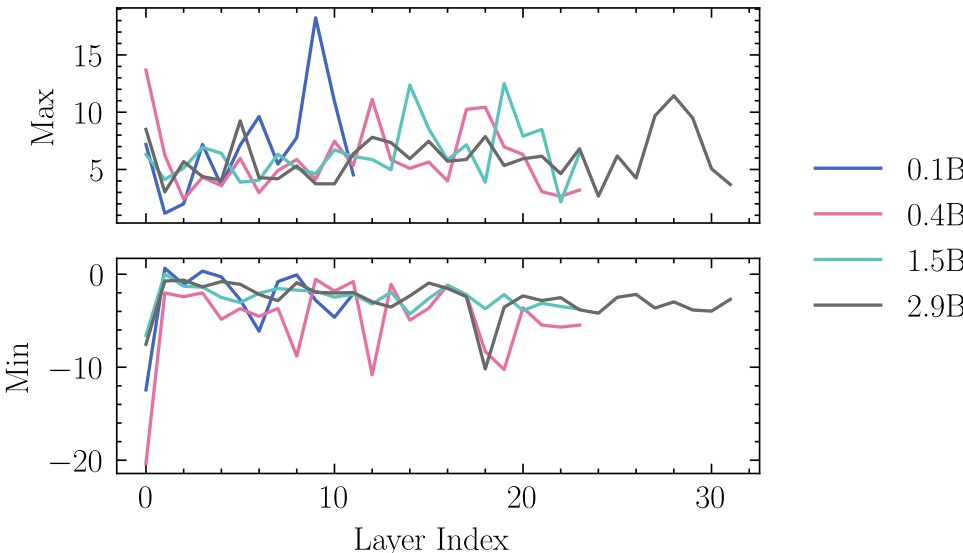

Figure 20: Maximum and minimum of $\alpha$ across layers in different models

consistency of the input and discovered that omitting the special token `<|endoftext|>` at the beginning of the input caused substantial and statistically significant differences in perplexity (PPL) and accuracy (ACC) for some RWKV-7 models.

Previous research highlights that Transformer models are sensitive to special tokens, and fine-tuning these tokens can yield notable improvements (Yang et al., 2023b). However, to our knowledge, no quantitative study has examined this effect systematically.

We analyzed the cases with the largest performance discrepancies, and identified a key pattern:

- The answer appears as the first word of the paragraph.
- This first word does not reappear elsewhere in the text.

This suggests that the model may struggle to retain the first token in memory.

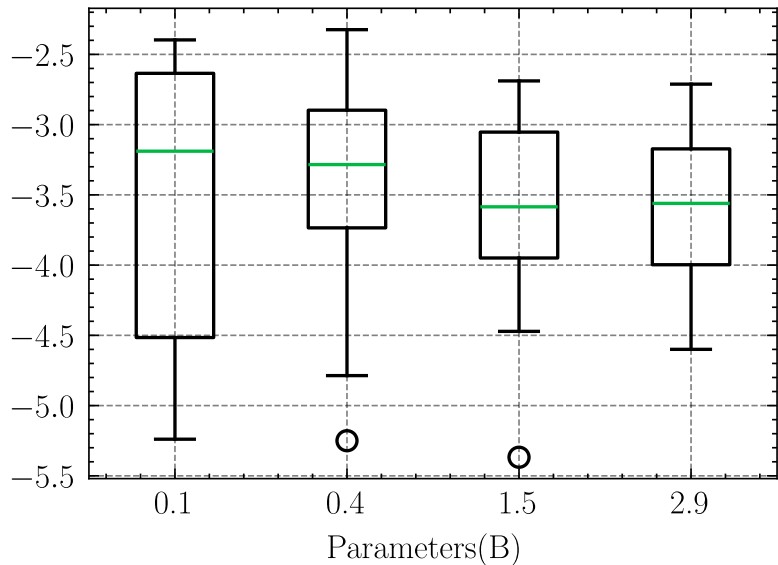

Figure 21: Box plots of biases of $d_t$ across models

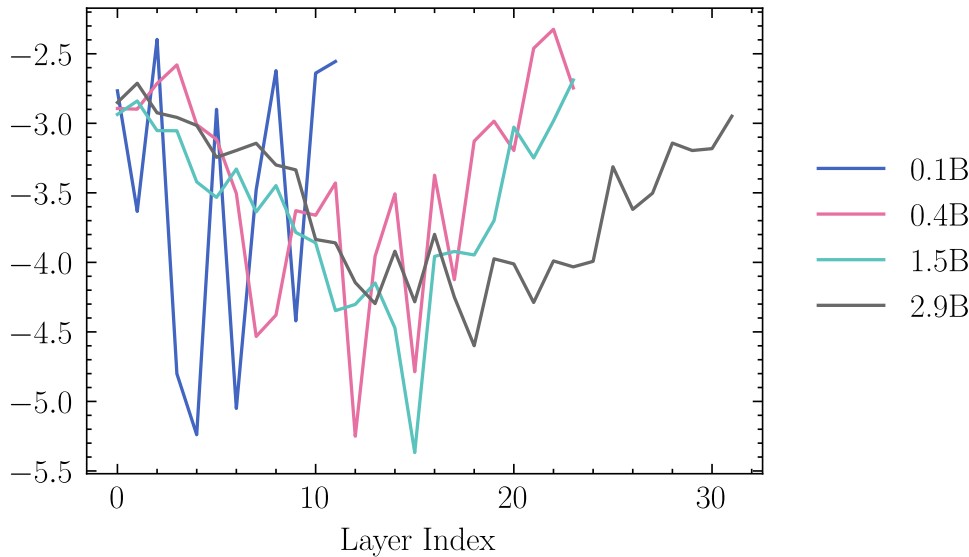

Figure 22: Mean biases of $d_t$ across layers for different models

In the LAMBADA test set, we identified 142 such examples out of a total of 5153 questions. One example is:

> Beth smoothed her wiry half-black, half-gray hair from her makeup-free face. In New Mexico, the natural look was common. Standing next to Cindy Fanucci, she felt like a disaster. She hid her ragged nails under the sleeves of her sweatshirt.
> "Hi, I'm Cindy. It's so nice to meet you, **Beth.**"

The following table summarizes the performance of different models with and without the padding token `<|endoftext|>`:

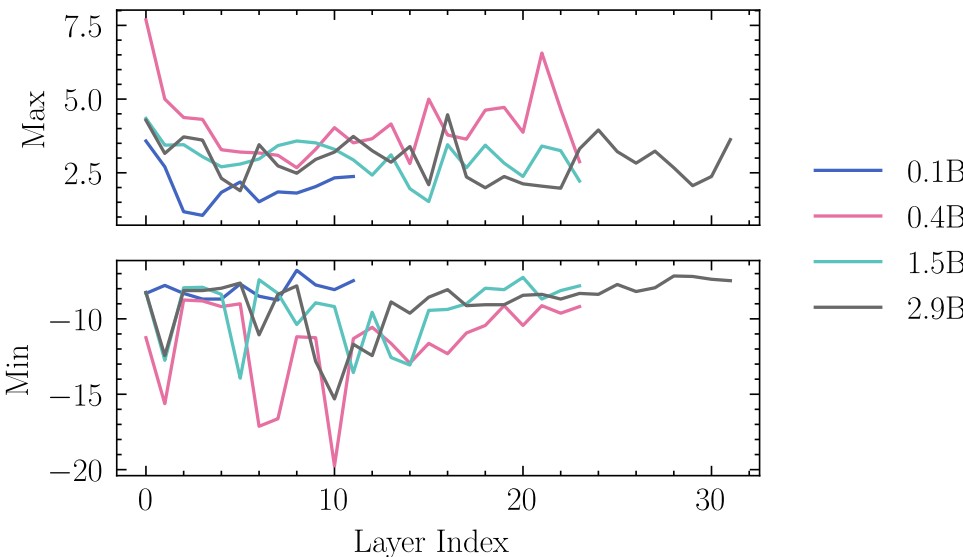

Figure 23: Maximum and minimum of biases of $d_t$ across layers in different models

| Model | EOS padding | PPL | ACC (%) | Significance |
|-------|-------------|-----|---------|--------------|
| RWKV7 World 0.1B | 0 | 357 | 9.2 | *** |
|  | 1 | 16.4 | 36.6 |  |
| RWKV7 World 0.4B | 0 | 42.7 | 28.9 | *** |
|  | 1 | 7.25 | 48.6 |  |
| SmolLM2 360M | 0 | 21.1 | 39.4 | * |
|  | 1 | 9.17 | 49.3 |  |
| Qwen2.5 0.5B | 0 | 12.2 | 47.9 | NS |
|  | 1 | 7.97 | 54.9 |  |

Table 21: Performance comparison of different models with and without the `<|endoftext|>` token in the partial set of 142 samples from LAMBADA. Significance levels: $^*\, p < 0.05$, $^{**}\, p < 0.01$, $^{***}\, p < 0.001$, NS = Not Significant.

The results indicate that the inclusion of the `<|endoftext|>` token improves the performance of RWKV-7 very significantly, especially for small models (e.g., RWKV-7 0.1B). This finding highlights the importance of proper state initialization and context setting for RNN-based architectures like RWKV. Unexpectedly, we also found that two consecutive `<|endoftext|>` tokens at the beginning can further improve the performance of RWKV-7, despite that `<|endoftext|>` never appears consecutively in the training corpus.

However, for Transformer-based models such as Qwen2.5-0.5B, we observe that the impact of the `<|endoftext|>` token is less pronounced, suggesting that these models may have better mechanisms for attending to the initial token.

As a result, for optimal performance when prompting RWKV-7, we recommend always including the `<|endoftext|>` token at the beginning of the prompt. For example, if you plan to use RWKV-7 as a chat assistant, consider the following structured prompt format:

```
<|endoftext|>User: <Your Question>

Assistant: <Assistant Answer>

User: <Another Question>

Assistant:
```

## Q   Limitations

Despite its strengths, the RWKV-7 architecture and models face certain limitations yet to be mitigated in future work.

**Numerical Precision.**   We observed that some operators, particularly the WKV7 kernel, are sensitive to the numerical precision of the implementation. This highlights the need for careful handling of numerical precision during model deployment. We also observed differences in training dynamics when using different kernels, which implies that the correct handling of precision while calculating and applying state updates is of utmost importance in this architecture.

**Lack of Instruction Tuning and Alignment.**   All RWKV-7 models presented in this work are pretrained base models and have not undergone the phase of Supervised Fine-Tuning (SFT) for instruction following nor alignment with human preferences (RLHF). Future efforts should focus on incorporating these capabilities to enhance the model's usability in real-world applications.

**Prompt Sensitivity.**   We found that the absence of the special token `<|endoftext|>` results in degraded performance of RWKV-7 models, e.g. inability to remember the first token of the input. See Appendix P for details.

**Compute Resources.**   Due to computational budget constraints, our training was limited on at most $12 \times 8 = 96$ Nvidia H800 GPUs. This falls short of the resources required for recent large-scale training efforts, such as DeepSeek-V3 (DeepSeek-AI et al., 2025). Additionally, we are forced to continue training from pre-existing checkpoints of earlier RWKV architectures and therefore re-use some parts of our dataset. This may limit the capabilities of our models versus pretraining from scratch. Scaling up RWKV-7 to larger sizes and datasets will require additional computational resources.

## R   Future Work

In addition to training larger RWKV-7 models with more tokens in the future, we also aim to explore several promising directions to further enhance the architecture and its capabilities.

**Speedup Techniques.**   A variety of speed optimization techniques were highlighted in the technical report of DeepSeek-V3 (DeepSeek-AI et al., 2025), including Dual Pipelining Mechanism, Mixture-of-Experts, Multi-Token Prediction, and FP8 Training. We are aware that many of these techniques are orthogonal to RWKV-7's architectural optimizations, therefore could be integrated to further accelerate training in later RWKV models. However, RWKV-7 has been trained completely without pipeline parallelism. We also noticed that there is room for speed optimization of RWKV-7 kernels and operators. We will explore both kernel-level optimizations and distributed training strategies in the future.

**Incorporating Chain-of-Thought Reasoning.**   We believe that RWKV-7, as a linear RNN, is well-suited for efficient Chain-of-Thought reasoning (Wei et al., 2022). However, this capability has been barely explored due to the lack of suitable reinforcement learning pipelines. In future work, we plan to incorporate deep thinking abilities into RWKV-7, enabling it to excel in tasks requiring multi-step logical reasoning and complex problem-solving.

