# OpenReview forum: "RWKV-7 "Goose" with Expressive Dynamic State Evolution"
_colmweb.org/COLM/2025/Conference — COLM 2025_

### Official Review · Reviewer_UGzf · 2025-05-11

**Rating:** 6
**Confidence:** 3
**Ethics Flag:** 1

**Summary:**

This paper presents RWKV-7 "Goose", a sequence modeling architecture described as having constant memory usage and constant inference time per token. The architecture introduces a generalized formulation of the delta rule with vector-valued gating and in-context learning rates, as well as a relaxed value replacement rule. Regarding empirical results, the 2.9 billion parameter model, stated to be trained on fewer tokens than other top models, achieves a new 3B state-of-the-art (SoTA) on multilingual tasks and matches the current 3B SoTA on English language downstream performance.

**Questions To Authors:**

1.	Could the dataset proposed RWKV-world3 dataset be used on other architectures? What would be the results?

2.	Figure 3 could be improved with better resolution and alignment.

**Reasons To Accept:**

1.	The architecture design of RWKV-7 looks nice and solid.

2.	The newly proposed RWKV World v3 dataset could be beneficial to the large language model pretraining community.

3.	Results shows that RWKV-7 shows comparable results with other transformer-based models while achieving better efficiency.

**Reasons To Reject:**

1.	One possible concern of the contribution of this paper is the entangled effect of the new model structure design and the newly proposed dataset, that is, the improvement and the comparable results of RWKV 7 could be gone if other models use the same dataset for their pretraining process. There lack ablation study experiments. This is especially important for the claims related to the multilingual performance.

2.	There lacks the evaluation on real-world generation tasks (e.g., story generation, stylist generation, role playing, etc.) which requires annotations from human evaluators.

---

> ### Author Response · Authors · 2025-06-01
>
> > 1. One possible concern ...
>
> We agree about the importance of separating out entangled effects. To address this specific issue we provided a set of ablation studies in which we train all models on the Pile. Please refer to Appendix L and Table 19 for these ablation study details. A concise plot of some of these results for RWKV-7 and Mamba-2 can also be found in the Pile models shown in Figure 7.
>
> > 2. There lacks the evaluation on real-world generation tasks (e.g., story generation, stylist generation, role playing, etc.) which requires annotations from human evaluators.
>
> While we agree that human evaluations and annotations are valuable for real-world generation tasks, they are unfortunately beyond the scope of our research focusing on developing new modeling architectures.
>
>
>
> > 1. Could the dataset proposed RWKV-world3 dataset be used on other architectures? What would be the results?
>
> This would be a fascinating study, and we would love to see other models adopt the multilingual RWKV-World3 dataset. Our belief is that it would be beneficial for any architecture, and could help improve accessibility for worldwide adoption of open source models. We have more open datasets coming soon that perform even better in our internal testing, and hopefully these will be a valuable resource to other model authors.
>
> > 2. Figure 3 could be improved with better resolution and alignment.
>
> Thank you for noticing this oversight. We have updated the image in the manuscript accordingly for camera ready.

---

### Official Review · Reviewer_QvXt · 2025-05-12

**Rating:** 8
**Confidence:** 4
**Ethics Flag:** 1

**Summary:**

This paper introduces RWKV-7 "Goose," a novel sequence modeling architecture that builds on past works on SSMs and Linear Attention and has constant memory usage during inference. In addition to the architectural innovations, the paper also introduces a new 3.1 trillion token multilingual corpus (RWKV World v3). Despite being trained on fewer tokens than comparable models, the 2.9B parameter RWKV7 model trained on the new corpus achieves state-of-the-art performance on multilingual tasks at its size and matches current SOTA on English benchmarks.

**Questions To Authors:**

1. When comparing architectures trained on identical datasets (as in the Pile experiments), what are the specific speed advantages of RWKV-7 over simpler architectures like Mamba2? Are there inference speed or memory usage tradeoffs that would help readers understand when to prefer RWKV-7?

**Reasons To Accept:**

1. The paper presents a novel RNN-based architecture building on past works on SSMs and Linear Attention. The model has linear complexity in the input length and constant memory usage during inference.
2. When compared to other models such as Mamba2 trained on Pile under comparable compute settings, the proposed RWKV-7 architecture reports improved performance.
3. The authors introduce a public 3.1 trillion token multilingual corpus. The RWKV-7 models trained on this corpus demonstrate impressive performance on multilingual benchmarks, achieving SOTA for its parameter count compared to larger models trained on more data. The model outperforms established architectures like Llama-3.2, Qwen2.5, and SmolLM2 on multilingual tasks with fewer training tokens.
4. The authors provide extensive benchmarking across standard NLP tasks, recent internet data evaluation, image understanding and specialized tests for associative recall and state tracking abilities.
5. The authors publicly release the pre-trained models and the training corpus.

**Reasons To Reject:**

1. The state update mechanism and overall architecture are quite complex, making the model potentially difficult to implement and understand for those not already familiar with linear attention variants.
2. While some ablations are provided, a more comprehensive analysis separating the contributions of each architectural component would strengthen the paper.

---

> ### Author Response · Authors · 2025-06-01
>
> > 1. The state update mechanism and overall architecture are quite complex, making the model potentially difficult to implement and understand for those not already familiar with linear attention variants.
>
> Although we cannot simplify the actual architecture further without reducing performance, we may be able to provide a more succinct description. Please let us know if you think it would be helpful for us to include the following 30 line numpy implementation in an appendix:
> ```python
> group_norm = lambda x, w, b : ((x - x.mean(axis=1,keepdims=1)) / (x.var(axis=1,keepdims=1) + 64e-5)**0.5).flatten() * w + b
> sigmoid = lambda x : 1/(1+np.exp(-x))
> def time_mixing(x, v0, last_x, S, params):
>     mr,mw,mk,mv,ma,mg, w_bias, r_k, Ww1,Ww2, Wa1,Wa2,a_bias, Wg1,Wg2 = params[:15]
>     k_k,k_a, Wr,Wk,Wv,Wo, ln_w,ln_b = params[-8:]
>
>     xr,xw,xk,xv,xa,xg = [x + m * (last_x - x) for m in [mr,mw,mk,mv,ma,mg]]
>
>     r = Wr @ xr
>     w = np.exp(-sigmoid(np.tanh(xw @ Ww1) @ Ww2 + w_bias)/np.e**0.5)
>     k = Wk @ xk
>     v = Wv @ xv
>     if v0 is None:
>         v0 = v
>     else:
>         Wv2,Wv1,v_bias = params[15:18]
>         v += (v0 - v) * sigmoid(xv @ Wv1 @ Wv2 + v_bias)
>     a = sigmoid(xa @ Wa1 @ Wa2 + a_bias)
>     g = sigmoid(xg @ Wg1) @ Wg2
>     kk = k * k_k
>     k += k * (a-1) * k_a
>
>     r,w,k,v,kk,a,r_k = [i.reshape(N_HEAD, HEAD_SIZE, 1) for i in [r,w,k,v,kk,a,r_k]]
>     kk /= np.maximum(np.linalg.norm(kk, axis=1,keepdims=1), 1e-12)
>
>     S = S * w.mT - S @ kk * (kk*a).mT + v * k.mT
>     y = S @ r
>
>     y = group_norm(y, ln_w, ln_b)
>     y += ((r * k * r_k).sum(axis=1,keepdims=1) * v).flatten()
>     return Wo @ (y * g), v0, x, S
> ```
>
> > 2. While some ablations are provided, a more comprehensive analysis separating the contributions of each architectural component would strengthen the paper.
>
> Thank you for bringing this to our attention. We will consider if there are other ablations we can run that might help illustrate the specific role that each component plays.
>
>
> > ### Questions To Authors:
> > When comparing architectures trained on identical datasets (as in the Pile experiments), what are the specific speed
>
> Training optimization can vary widely across different model architectures, sizes, and devices, so such comparisons may unfortunately be beyond the scope of this paper.

---

> > ### Comment · Reviewer_QvXt · 2025-06-02
> > **Response to the authors**
> >
> > Thank you for the additional clarifications - I am maintaining my score.

---

### Official Review · Reviewer_GjhW · 2025-05-13

**Rating:** 8
**Confidence:** 4
**Ethics Flag:** 1

**Summary:**

The paper introduces RWKV-7 "Goose," a new RNN architecture for effective language modeling.  RWKV-7 builds on recent work for the delta rule, thus offering new modeling capabilities not featured in many recent SOTA recurrent models (e.g., Mamba/Mamba-2, RWKV-6, etc.).  Along with this new architecture, the paper also debuts a large-scale (3.1 trillion tokens) open-source corpus supporting English, math, and multilingual task modeling.  A large portion of the paper is dedicated to flushing out the design choices, situating these choices within the context of previous work, and exactly highlighting what contributions are novel.  Finally, the paper ends with a large number of experiments, demonstrating the:
- speed and memory efficiency of the developed RWKV-7 for modern hardware
- performance on a large number of standard benchmarks, as well as multilingual benchmarks, demonstrating that RWKV-7 is highly competitive with SOTA attention and RNN LLMs of similar parameter counts (many of which are pre-trained on significantly many more tokens)
- RWKV-7 is SOTA/highly competitive with other models on the Mechanistic Architecture Design tasks

**Questions To Authors:**

- "we expand the in-context learning rate" <- What is the ICL rate?  Are the authors referring to the context window conditioned on (i.e., previous token for Mamba vs previous n tokens for autoregressive Transformers?).  Addendum after reading: please add a forward reference to where the ICL rate is defined, i.e., "we expand the in-context learning rate, described in Section..."

- "For now, we choose to allow only part of the range of possible negative eigenvalues in our pre-trained large language models due to experimentally observed training instabilities." <- Can the authors speak to both the range of values and the types of instabilities encountered?  Might these instabilities be side-stepped and the range of negative values increased during fine-tuning?

- What model is Flash Attention 3 being run on in Figure 3?

- Why is English Lambada perplexity not reported in Table 2?

- Unlikely that it would have placed highly in its weight	class, but it would have been interesting to see mamba-2.8b-slimpj in Table 2 and 3.

- To verify, the non-instruct-tuned versions of relevant models are used for comparison in Table 2 and 3,	correct?  If possible, could the authors include an appendix table of the competitor HF checkpoints?

- What are the parameter counts for the models in Table 4?

- Could you please add implementation details for the compression rate metric?  Also, to make the paper self-contained, could you add a description of the compression rate metric used in Table 15.

## Additional comments
- "Ashish Vaswani, Noam Shazeer, Niki Parmar, Jakob Uszkoreit, Llion Jones, Aidan N. Gomez, Lukasz Kaiser, and Illia Polosukhin. Attention is all you need, 2023." <- Attention citation is incorrect
- "For short sequences, much of this cost can be covered by modern GPU parallelism techniques" <- Ideal for Flash Attention citation
- "as a RNN" <- "as an RNN"

**Reasons To Accept:**

The paper is well written, and the pace and clarity of discussion are appreciated.  The architectural choices are intuitive, well explained, and improve the modeling expressiveness over other recent RNN innovations (which are reflected in the overall results).  The included benchmarks are extensive and comprehensive, and further demonstrate the efficacy of the design choices for RWKV-7.  In particular, though the model is pre-trained on far fewer total tokens than other SOTA attention/RNN models, it remains competitive across a large number of natural language english/multilingual tasks.  Thus, the RWKV-7 design, RWKV World v3 corpus, and actual pre-trained model checkpoints serve as valuable contributions to the community.  There are also significant additional experiments which show the RWKV-7 model's aptitude on recent synthetic tasks (stress testing several sequence modeling capabilities) and compression tasks.

**Reasons To Reject:**

There are only minor issues with clarity/missing details (see questions below).

---

> ### Author Response · Authors · 2025-06-01
>
> > "we expand the in-context learning rate" <- What is the ICL rate? Addendum after reading: please add a forward reference to where the ICL rate is defined, i.e., "we expand the in-context learning rate, described in Section..."
>
> Thank you for the suggestion. We have updated the manuscript to reflect this change for camera-ready.
>
> > Can the authors speak to both the range of values and the types of instabilities encountered? Might these instabilities be side-stepped and the range of negative values increased during fine-tuning?
>
> You raise a very good point, and we supply some details about this in Theorem 1 in Appendix B. It is worth noting that, although all the eigenvalues are bounded in $[-1,1]$, this does not always bound the largest singular value (the spectral norm) below one, which is a sufficient (but not necessary) condition for overall stability. In practice, in some timesteps the largest singular value does exceed 1, but further inspections (Appendix M) show that it does not affect general stability of state values. We will continue to investigate this issue in our future studies.
>
> > What model is Flash Attention 3 being run
>
>  In Figure 3 we show results for isolated kernels, rather than full models.
>
> > Why is English Lambada perplexity not reported
>
> Thank you for catching this oversight, which may have occurred due to concerns that the text in the table had gotten too small. We will add these results back into the paper for the camera ready.
> | Size | Name | Perplexity |
> |--|--|--|
> |0.1B|RWKV5-World1-0.1B | 22.8 |
> | | SmolLM2-135M | 19.3 |
> | | RWKV7-World2.8-0.1B | 12.6 |
> |0.4B| RWKV5-World2-0.4B | 8.9 |
> | | SmolLM2-360M | 9.4 |
> | | Qwen2.5-0.5B | 10.6 |
> | | RWKV7-World2.9-0.4B | 7.0 |
> |1.5B| RWKV6-World2.1-1.6B | 4.6 |
> | | Llama3.2-1B | 5.7 |
> | | SmolLM2-1.7B | 4.45 |
> | | Qwen2.5-1.5B | 5.7 |
> | | RWKV7-World3-1.5B | 4.17 |
> |3B| Mamba-2.8B-Slimpj | 5.9 |
> | | RWKV6-World2.1-3B | 3.86 |
> | | Llama3.2-3B | 3.94 |
> | | Qwen2.5-3B | 4.6 |
> | | RWKV7-World3-2.9B | 3.44 |
>
> > it would have been interesting to see mamba-2.8b-slimpj
>
> Sure! Here are the results:
> | Model | lmb.o(ppl) | lmb.o(acc) | hella(acc_n) | piqa | arcE | arcC | glue | WG | sciq | mmlu | avg |
> |-|-|-|-|-|-|-|-|-|-|-|-|
> | Mamba-2.8B-Slimpj | 5.9 | 64.4 | 71.0 | 77.1 | 72.6 | 38.8 | 50.3 | 65.9 | 91.3 | 26.4 | 62.0 |
>
> | Model | lmb.m(ppl) | lmb.m(acc) | pawsx | xcopa | xnli  | xsClz | xwin | avg |
> |-|-|-|-|-|-|-|-|-|
> | Mamba-2.8B-Slimpj | 38 | 44.0 | 54.6  | 53.7  | 38.8  | 54.8  | 73.5  | 53.2  |
>
> > To verify, the non-instruct-tuned versions of relevant models are used for comparison in Table 2 and 3, correct? If possible, could the authors include an appendix table of the competitor HF checkpoints?
>
> Thank you for the excellent suggestion - we have updated the tables in the manuscript to add hyperlinks to all listed models. All models listed in Table 2 and 3 are base models without instruction fine tuning.
>
> > What are the parameter counts for the models in Table 4?
>
> According to the default settings listed in https://github.com/athms/mad-lab/blob/main/benchmark.py, the models have two layers with dimension 128. This accounts for approximately 2.5M parameters.
>
> > Could you please add implementation details for the compression rate metric? Also, to make the paper self-contained, could you add a description of the compression rate metric used in Table 15.
>
> Thank you for the suggestion. We have added the following to the caption under Table 15 for camera-ready: We define compression rate as: $$ \frac{\log_2(e) * \text{average loss over document} * \text{document tokens}}{8 * \text{document length in bytes}}. $$ This definition is also consistent with Shannon’s information theory.
>
> We sincerely hope these clarifications could strengthen the manuscript. Please don’t hesitate to reach out if there are any further questions or suggestions.

---

> > ### Comment · Reviewer_GjhW · 2025-06-07
> > **Reply**
> >
> > The authors have done well to answer all my questions/comments.  I retain my score of clear accept.

---

### Official Review · Reviewer_bdD1 · 2025-05-16

**Rating:** 8
**Confidence:** 5
**Ethics Flag:** 1

**Summary:**

RWKV-7 Goose introduces a new sequence modeling architecture that achieves state-of-the-art performance at the 3B parameter scale for multilingual tasks while matching SoTA on English language tasks, despite being trained on fewer tokens than competing models.

The architecture features constant memory usage and inference time per token, making it more efficient than Transformer models for long sequences.

Key contributions includes:

- A generalized formulation of the delta rule with vector-valued gating and in-context learning rates

- A relaxed value replacement rule that allows selective state updating

- Improved state tracking capabilities that can recognize all regular languages

- Open-sourced models and recipes to the community.

The paper demonstrates that RWKV-7 has expressivity that exceeds TC complexity class under standard complexity conjectures, which is a theoretical advantage over Transformers.  But I am not so sure or confident about this conclusion, which I would encourage authors to elaborate further.

**Questions To Authors:**

In my view, the paper would benefit from:

Moving some of the key theoretical results from the appendix to the main paper

Providing more intuitive explanations of why these complexity results matter

Including concrete examples that demonstrate how these theoretical capabilities translate to practical advantages

Better connecting the theoretical guarantees to the empirical results observed in benchmarks

**Reasons To Accept:**

- Novel Architecture: The paper presents a significant evolution of linear RNN-based architectures through the generalized delta rule, offering a compelling alternative to Transformers with linear complexity.

- Strong Empirical Results: RWKV-7 achieves state-of-the-art performance on multilingual tasks at 3B scale despite being trained on fewer tokens than competitors, demonstrating impressive parameter efficiency.

- Computational Efficiency: The architecture offers constant memory usage and inference time per token, addressing key limitations of Transformers for long sequences.

- Open Source Contribution: The research provides multiple pre-trained models and a large multilingual corpus, contributing valuable resources to the research community.

I also want to highlight their comprehensive evaluation: The paper includes extensive benchmarking across English and multilingual tasks, long-context understanding, associative recall, and state tracking capabilities, setting good examples for future model architecture design papers for scientific studies.

**Reasons To Reject:**

- Numerical Precision Issues: The paper notes sensitivity to numerical precision in certain operators, particularly the WKV7 kernel, which may complicate deployment. But I don't think this belies the strength of the paper.

- Lack of Instruction Tuning: The presented models are base models without instruction tuning or alignment, limiting direct comparability to fully-tuned models in practical applications. But this would be helpful to further understand how this model can be actually used in the production environment.

---

> ### Author Response · Authors · 2025-06-01
>
> > The paper demonstrates that RWKV-7 has expressivity that exceeds TC complexity class under standard complexity conjectures, which is a theoretical advantage over Transformers. But I am not so sure or confident about this conclusion, which I would encourage authors to elaborate further.
>
> Thank you for this point about needing further elaboration on the actual advantages that come from the complexity class improvements of RWKV-7 over Transformers. Transformers aggregate history through commutative sums, which limits their ability to solve sequential state tracking problems where the precise order of actions matters for the result (formally, these problems are outside TC0 under standard conjectures). A key example of state tracking is computing the state of a chess board given a sequence of moves, each specified in notation like ‘e2e4’, indicating the source and target square. Transformers, S4 and Mamba cannot solve this chess state tracking problem beyond short input lengths unless extra 'thinking' tokens are given (See https://arxiv.org/abs/2404.08819 and https://arxiv.org/abs/2310.07923). In contrast, our result shows that RWKV-7 can solve the chess state tracking task for any sequence length as long as the model has at least four layers with sufficient width, and a head size that is linear with respect to the number of board positions. We prove this result theoretically in Appendix D. We will add further description and examples like this for camera-ready.
>
> > Numerical Precision Issues: The paper notes sensitivity to numerical precision in certain operators, particularly the WKV7 kernel, which may complicate deployment. But I don't think this belies the strength of the paper.
>
> Thank you for raising this important concern about numerical precision. Currently, we have multiple implementations of WKV7 kernel for model training, some written in CUDA and some in Triton, some optimized for higher precision and some for balanced performance under different sequence lengths. For example, a "backstepping" implementation computes gradients in FP32, achieving a numerical error of 9e-5, which is generally satisfactory. On the other hand, a faster "chunked" implementation uses BF16 precision, reducing computation time by 2x (saving ~8ms per pass in our 0.1B model) but introduces a slightly higher error of 5e-3. Unlike Flash Attention v3, our kernels have not yet undergone GPU architecture-specific optimizations (like Nvidia Hopper or AMD MI300 series), and we are actively working to optimize these implementations further.
>
> > Lack of Instruction Tuning: The presented models are base models without instruction tuning or alignment, limiting direct comparability to fully-tuned models in practical applications. But this would be helpful to further understand how this model can be actually used in the production environment.
>
> Our pre-trained RWKV-7 base models do not feature any true instruction tuning or alignment, having been exposed to only the limited amount of open-source instruction data in the pretraining stage contained in the RWKV World v3 corpus. Specific instruction tuning and alignment processes can vary widely, and are unfortunately beyond the scope of our architecture paper.
>
> > Questions To Authors:
> > In my view, the paper would benefit from:
> > Moving some of the key theoretical results from the appendix to the main paper
> > Providing more intuitive explanations of why these complexity results matter
> > Including concrete examples that demonstrate how these theoretical capabilities translate to practical advantages
> > Better connecting the theoretical guarantees to the empirical results observed in benchmarks
>
> These are great ideas and seem feasible to implement. While we were limited by space concerns for the submission version, we hope that the camera ready might allow us one more page that we can use for this purpose. We can give examples, including experiments shown in the paper, of the kinds of problems these complexity improvements allow the model to solve without requiring extra tokens.

---

> > ### Comment · Reviewer_bdD1 · 2025-06-01
> > **response**
> >
> > Thank authors for their comments, and I remain positive about the paper!

---

### Decision · Program_Chairs · 2025-07-08

**Decision:**

Accept

**Comment:**

The paper proposes a novel architecture for language modeling called RWKV-7, based on RNNs and yet outperforming similarly-sized transformer and Mamba models on multilingual tasks and showing promise on English tasks. The authors will release their model and the 3.1-trillion token data used to pre-train the 2.9-billion parameter variant.

The reviewers were enthusiastic about the paper, apparently convinced by the strong experimental setup and the impressive results. Minor qualms were answered to their satisfaction; one reviewer who was less supportive asked a more fundamental question but did not engage with the reply which, to me, made sense. It seems like there is no serious concern about accepting this rigorous, useful piece of work.

[Automatically added comment] At least one review was discounted during the decision process due to quality]